



# A risk assessment framework for interacting tipping elements

Jacques Bara[1,2], Nico Wunderling[3,4,5], and Wolfram Barfuss[1,2,4,6]

[1]Center for Development Research, University of Bonn, 53113 Bonn, Germany
[2]Transdisciplinary Research Area Sustainable Futures, University of Bonn, 53115 Bonn, Germany
[3]Center for Critical Computational Studies, Goethe-University Frankfurt, 60322 Frankfurt am Main, Germany
[4]Earth Resilience Science Unit, Potsdam Institute for Climate Impact Research (PIK), Member of the Leibniz Association, 14412 Potsdam, Germany
[5]Senckenberg Research Institute and Natural History Museum, Member of the Leibniz Association, 60325 Frankfurt am Main, Germany
[6]Institute for Food & Resource Economics, University of Bonn, 53115 Bonn, Germany

**Correspondence:** Jacques Bara (jbara@uni-bonn.de) and Wolfram Barfuss (wbarfuss@uni-bonn.de)

**Abstract.** Tipping elements, such as the Greenland Ice Sheet, the Atlantic meridional ocean circulation (AMOC) or the Amazon rainforest, interact with one another and with other non-linear systems such as the El-Nino Southern Oscillation (ENSO). In doing so the risk of any one element collapsing into a degraded state can be drastically affected, typically increasing due to the interactions. In this work, therefore, we propose a fully probabilistic network model for risk assessment of interacting tipping elements that coherently incorporates literature-based belief assessments of intra-element interactions. We provide analytic results for the equilibrium risks of nine interacting tipping elements, the existence and stability of their stationary distributions and convergence times to the equilibrium solution. Moreover we simulate their tipping risks until 2350 using emission pathways from the shared socio-economic pathways (SSP 1-1.9, 1-2.6, 2-4.5, 3-7.0, and 5-8.5). Compared to the hypothetical no-interactions case, we find that interactions tend to destabilise the climate system, for instance the coral reefs are likely to have collapsed by 2100 even under the most optimistic scenario (SSP1-1.9). The effects of interactions, however, are most noticeable after 2100, especially for the highest shared socio-economic pathways (SSP3-7.0 and SSP5-8.5). In summary, our comprehensive risk assessment framework for tipping elements indicates that rapid mitigation is essential to keep temperatures as close as possible to $1.5°C$ in the short term and below $1°C$ in the longer run.

## 1 Introduction

The dynamics of Earth systems often show non-linearity, path dependency and chaotic behaviour (Burke et al., 2018; Westerhold et al., 2020). The West Antarctic Ice Sheet (WAIS), for example, is a global climate tipping element (TE) (Armstrong McKay et al., 2022) which may undergo a self-sustaining collapse due to marine ice sheet instability (Feldmann and Levermann, 2015; Waibel et al., 2018) should it pass its tipping point. Merely observing and assessing the state of one can be a challenging endeavour – WAIS evolves over hundreds to thousands of years (Van Breedam et al., 2020; Armstrong McKay et al., 2022) and there is often insufficient data to cover the entirety of its life cycle. Moreover, to understand one tipping element





is to grasp its many components across a hierarchy of levels and scales, using various perspectives and often inter-disciplinary approaches (Wimsatt, 1994).

Furthermore, tipping points are not independent of one another; they can stabilise and destabilise other ones, causing cascading effects that ripple throughout the entire Earth system (Rocha et al., 2018; Steffen et al., 2018; Wunderling et al., 2024). In

general, most interactions destabilise other tipping elements and the Earth system as a whole (Wunderling et al., 2023, 2024). Even at the local or regional levels, interacting regime shifts facilitate cascades (Rocha et al., 2018). However, there are some notable exceptions. For example, Sinet et al. (2023, 2024) found that the polar ice caps may stabilise the Atlantic meridional ocean circulation (AMOC) through saltwater flux contingent upon the rates of and delays between their collapse. Reciprocally, Jackson et al. (2015) found that a weakening AMOC may stabilise the Greenland ice sheet, although this has been contested

(Wunderling et al., 2021a, 2023).

The Earth system and tipping points are epistemically complex: containing both high and deep uncertainty, which are different epistemic situations used to distinguish the extent to which we know or do not know about a topic (Lam and Majszak, 2022). In high uncertainty, the state of knowledge can be either more or less incomplete/inadequate, compared to other cases. Whereas in deep uncertainty, such comparisons cannot even be made as there is little to no consensus amongst experts on the

appropriate alternative outcomes/scenarios or even system model. Given the unprecedented severity and frequency of extreme Earth system events, there have been increasing calls on policymakers and global leaders to adopt a global risk management approach, such as Planetary Solvency (Trust et al., 2024, 2025), that seriously considers such deep uncertainty.

Overcoming both uncertainties has often required expert judgement to go beyond the outputs of climate modelling (Lam and Majszak, 2022), exemplified best by reports from the Intergovernmental Panel on Climate Change (IPCC), such as the most

recent assessment report (AR6) (Lee et al., 2021). First, events that are not captured well by current complex Earth system models can be evaluated and incorporated through expert judgement. Second, through their assessment, an expert is able to coalesce large parts of the literature over multiple models and results, thus overcoming some of the deep uncertainty. Third, through synthesising and condensing large parts of the literature together, this serves as an essential communication tool for decision-makers and other non-experts to understand the research. Finally, by relying on rigorous elicitation protocols, this

approach can be systematised across many experts in multiple fields.

In the literature on interacting tipping elements, the frequently cited seminal work by Kriegler et al. (2009) is precisely an expert elicitation seeking to characterise/quantify the individual tipping points as well as to bound the conditional probability – what the authors call a probability ratio or $PF$ – of one element being triggered given that another has already tipped. For example, after systematic weighting, the experts judged that if WAIS has already deteriorated, then the probability for the

Greenland Ice Sheet (GIS) to trigger increases by a factor of 1 to 2. This systematic elicitation has then informed several of the past tipping point assessments (Wunderling et al., 2021a); in qualitative Boolean models (Gaucherel and Moron, 2017); in a conceptual mechanistic model (Sinet et al., 2023); and in dynamical systems models (Cai et al., 2016; Wunderling et al., 2020, 2021a, 2023) where it often has informed model calibration and parametrisation of interactions.

These modelling approaches, however, are limited regarding their treatment of uncertainty. First, uncertainty has only been

incorporated in the form of stochastic noise – for example noise-induced tipping (N-tipping) in some climate models occur





when "noisy fluctuations result in the system departing from a neighbourhood of a quasi-static attractor" (Ashwin et al., 2012) – or through confidence intervals on parameter values – which is often accounted for via computationally heavy Monte Carlo simulations such as in (Wunderling et al., 2021a, b, 2023; Rosser et al., 2024). Second, although other works have modelled individual TEs probabilistically (Barfuss et al., 2018, 2020) they neglect the interactions between tipping elements which may cause a cascade of tipping events. Third, even in dynamical models that do consider the TE-TE interactions (Wunderling et al., 2020, 2021a, b, 2023), these were implemented as linearly additive coupling terms in the evolution of some state variable or observable, whereas the pairwise PFs of Kriegler et al. (2009) are multiplicative factors that act on the probability of a TE being tipped.

We remedy these shortcomings by providing a risk assessment modelling framework of interacting tipping elements with a more coherent treatment of uncertainty, which is also computationally more efficient. We take the perspective that the environment itself is probabilistic insofar as the 'real' state of the system is indirectly measured/observed, due to its complex nature and even its conceptualisation. We thus model each tipping element as a 2-state Markov chain with a Prosperous state and a Degraded state, similarly to other probabilistic approaches (Barfuss et al., 2018, 2020, 2024), whose transitions, in contrast to those previous models, further depend on the state of their network neighbours. In this way, though there is a latent state for each tipping element to be in, what is modelled/observed is the probability to be in, say, the Degraded state. This allows us to greatly reduce the dimensions of the Monte Carlo ensembles requiring, therefore, far less heavy computation.

Moreover, our approach inherently allows for conditional interactions, that is the probability for one tipping element to be triggered is increased/decreased when another has tipped, more coherently incorporating the probability factors derived through expert elicitation (Kriegler et al., 2009), unlike the linearly additive interaction terms in other models (Wunderling et al., 2020, 2021a, b, 2023). Finally, by taking a probabilistic approach, we soften their assumption that "a state change is initiated as soon as the increase in GMT exceeds the critical temperature" (Wunderling et al., 2021a). We do so through a tipping belief logistic function, which accounts for the physical sensitivity of the element, or at least uncertainty in the critical temperature. Furthermore, through this temperature dependence we incorporate the anthropogenic drivers and effects on environmental dynamics, which are mediated by global warming, using the SSP scenarios from the IPCC's Sixth Assessment Report (Lee et al., 2021) for analysis of the short-term risk and a range of fixed levels for the equilibrium risk.

In Sect. 2 we introduce our probabilistic approach to modelling interacting tipping elements: first, we set up the mathematical framework for a network of interacting Markov chains (Sects. 2.1 and 2.2) including under external forcing due to global warming (Sect. 2.3); second, in Sect. 2.4 we give an overview of the nine specific tipping elements we will consider including their inferred parameters and their pairwise interactions which is further illustrated in Fig 1. Having established the model and parametrisation thereof, in Sect. 3.1 we present the theoretical analysis regarding the equilibrium degraded probabilities (Sect. 3.1.1), their stability (Sect. 3.1.2) and the convergence time of long-term solutions to the equilibrium (Sect. 3.1.3). We evaluate the equilibrium solution for the interacting network of tipping elements in Sect. 3.2 and provide computational solutions for their short-term evolution under various SSP scenarios in Sect. 3.3. We conclude with a discussion of our results in Sect. 4.





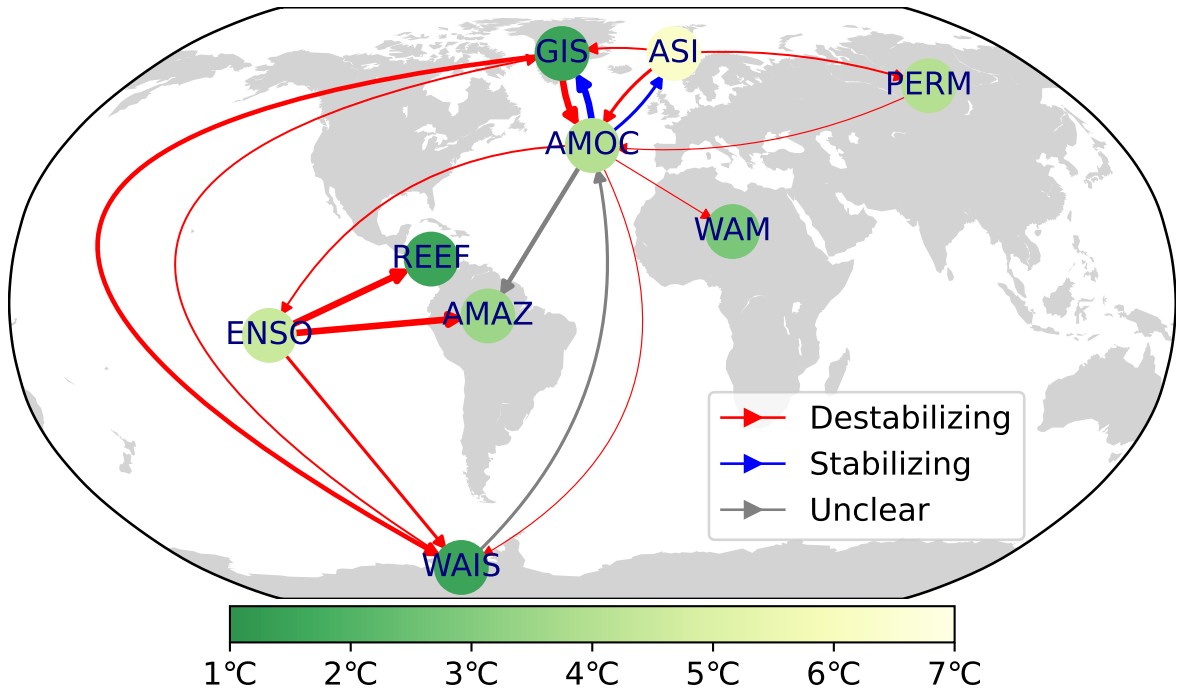

**Figure 1.** Illustration of the tipping element interaction network, with data from Armstrong McKay et al. (2022) and Wunderling et al. (2024). Nodes represent individual tipping elements with their colours indicating the mean of the respective tipping threshold temperature. Node locations are not exhaustively representative but have instead been chosen for illustrative purposes; for example, low-latitude coral reefs are also found around the islands of Indonesia and off the east coast of Africa. Moreover, note that, although its status as a tipping element is debated (Armstrong McKay et al., 2022), we include the El-Niño Southern Oscillation (ENSO) due to its important feedbacks on both global core and regional tipping elements. Edges indicate interactions due to one tipping element on another, with destabilizing edges in red, stabilizing in blue, and unclear/mixed in gray. Edge thickness indicates the maximum value of the signed edge weight, thus thicker edges denote potentially stronger interactions.

## 2 Methods

### 2.1 Two-State Markov chains preliminary

In general, for a 2-state Markov chain (MC) with transition rate $p$ to go from state $S_0$ to $S_1$ – and transition rate $q$ from $S_1$ to $S_0$ – the transition matrix $\mathbf{T} = (T_{SS'} : S, S' \in \{S_0, S_1\})$ is given by Eq. (1). The update equation for the probability to be in one of the states, say state $S_1$, at time $t+1$, $\mathbb{P}_{t+1}(S_1)$, can then be found in terms of $p, q$ and the probability at time $t$ given by Eq. (2). The dynamics of the whole system are sufficiently captured by the dynamics of one state since we have $\mathbb{P}(S_0) + \mathbb{P}(S_1) \equiv 1$. Given an initial distribution of $\mathbb{P}_0(S_1)$ at time $t = 0$, the explicit time evolution at time $t$ is given by Eq. (3). At equilibrium,



$\mathbb{P}_{t+1}(S_1) = \mathbb{P}_t(S_1)$, giving the stationary distribution, $\mathbb{P}_*(S_1)$, in Eq. (4) which, assuming fixed $p, q$, is stable if $p + q > 0$.

$$\mathbf{T} = \begin{pmatrix} 1-p & p \\ q & 1-q \end{pmatrix} \tag{1}$$

$$\mathbb{P}_{t+1}(S_1) = p(1 - \mathbb{P}_t(S_1)) + (1-q)\mathbb{P}_t(S_1) \tag{2}$$

$$\mathbb{P}_t(S_1) = \frac{p}{p+q}(1 - (1-p-q)^{t-1}) + (1-p-q)^t \mathbb{P}_0(S_1) \tag{3}$$

$$\mathbb{P}_*(S_1) = \frac{p}{p+q} \tag{4}$$

To find the stability of Eq. (4) corresponds to finding the absolute value of the Jacobian, which, in this case, is simply the derivative of Eq. (2). Formally, for $x = \mathbb{P}_t(S_1)$ and a map $f : [0,1] \to [0,1]$ such that $\mathbb{P}_{t+1}(S_1) = f(x)$, the stationary distribution $x_* = \mathbb{P}_*(S_1)$ is stable if $|f'(x_*)| < 1$. From Eq. (2), $f'(x) = 1 - (p+q)$ such that the stability condition simply becomes $0 < p + q < 2$.

## 2.2 Interacting Markov chains

For a set of Markov chains, $V$, let $G = \{V, E\}$ be a graph of pairwise interactions between individual Markov chains. A directed edge $(i,j) \in E$ exists if there is a causal link or effect due to MC $i \in V$ on another MC $j \in V$, disallowing self-loops. The element $a_{ij}$ of adjacency matrix $\mathbf{A} = (a_{ij} : \forall i, j \in V)$ corresponds to the probability factor $PF(i \to j)$ due to one tipping element $i$ being triggered on another element $j$. In general the edge from $i$ to $j$ is represented by $a_{ij}$ which may be weighted – indicating some positively valued coupling strength – and signed – indicating the polarity of the effect (e.g. stabilising or destabilising). For signed edges, we denote the adjacency matrix of destabilising edges with $\mathbf{A}^+ = (a_{ij}^+ : \forall i, j \in V)$ with with $a_{ij}^+ = PF(i \to j)$ when $PF(i \to j) > 1$. Similarly, we denote $\mathbf{A}^- = (a_{ij}^- : \forall i, j \in V)$ all the stabilising edges, i.e. those with $PF(i \to j) < 1$ has a corresponding adjacency element $a_{ij}^- = PF(i \to j)^{-1}$.

The state of each MC $i \in V$ is denoted by $\mathcal{S}_i$ which can take one of two environmental states – either Prosperous $\mathcal{P}$ or Degraded $\mathcal{D}$ – and has an associated internal collapse probability $p_{c,i} \in (0,1)$ and recovery probability $p_{r,i} \in (0,1)$, which are the Markov transition probabilities between the possible environmental states due to the internal processes alone. Moreover, let the probability of node $i$ being in the Degraded state be denoted $D_i \equiv \mathbb{P}(\mathcal{S}_i = \mathcal{D})$. As probability factors are conditional upon one element having already been triggered, then, in general, degraded destabilising in-neighbours increase the collapse probability, while stabilising ones decrease the probability to collapse. We therefore modify the individual collapse probability by the interactions due to all in-neighbours into an effective probability to collapse $\tilde{p}_{c,i}$.

$$\tilde{p}_{c,i} = p_{c,i} \frac{1 + \sum_{j \in V} a_{ji}^+ D_j}{1 + \sum_{j \in V} a_{ji}^- D_j} \tag{5}$$

Each Markov chain evolves according to the transition matrix in Eq. (1) and the corresponding update equation in Eq. (2) with $p = \tilde{p}_{c,i}$ and $q = p_{r,i}$, however there is now a dependence on the states of its in-neighbours. As such an explicit expression for the time evolution of each node requires solving a set of non-linear coupled maps, such that a general solution is cumbersome



to write down as a closed form expression. Instead, as in Eq. (2), we derive the update equation $D_i[t+1]$ in Eq. (6). Note for notational brevity, where relevant, all terms in the right hand side of the equation are evaluated at time $t$.

$$D_i[t+1] = \frac{1 + \sum_{j \in V} a_{ji}^+ D_j}{1 + \sum_{j \in V} a_{ji}^- D_j} p_{c,i}(1 - D_i) + (1 - p_{r,i})D_i \tag{6}$$

### 2.3 Anthropogenic collapse and recovery

The dynamics of nature are not purely driven by natural/physical processes but also by the actions of individuals, stakeholders and institutions. We consider the anthropogenic effects as being mediated by global warming. That is, as Armstrong McKay et al. (2022) finds, when the global mean surface temperature (GMST) exceeds the critical threshold of an individual tipping element, that element is far more likely to be triggered and thus begin its timely collapse. To that end, for each tipping element $i$, we do a similar decomposition into an internal natural process and a GMST-driven anthropogenic process. In particular, given

a GMST anomaly relative to pre-industrial times of $\Delta T$, for tipping element $i$ with timescales $\tau_{c,i}$ and $\tau_{r,i}$ and an estimated critical temperature threshold of $T_{lim,i}$ we have that the internal probabilities to recover and collapse are given by,

$$p_{r,i} = \frac{1}{(\tau_{r,i}/\tau_m) + 1}, \qquad p_{c,i} = \frac{1}{(\tau_{c,i}/\tau_m) + 1} \cdot \frac{1}{1 + \exp\left(-\beta_i(\Delta T - T_{lim,i})\right)} \tag{7}$$

where $\tau_m$ is the duration of a model time step in time units – unless otherwise stated we henceforth set $\tau_m = 1$ year – and $\beta_i$ is the sensitivity of the logistic function. One way to interpret this parameter is physically: if a tipping point with low $\beta_i$

has crossed its threshold by a small amount then its tipping probability is only marginally increased; whereas one with a high $\beta_i$ would experience a large increase in its tipping probability. Another way, as we do in this work, is to quantify the deep uncertainty amongst the expert assessments: that is when the range of plausible tipping thresholds is large, indicating the wide range in expert opinion, the sensitivity $\beta_i$ is low; whereas a small range in the threshold suggests a high degree of confidence and/or agreement and therefore a high $\beta_i$.

Notice that $\Delta T$ may in general evolve over some timescale $\tau_w$; the Anthropocene has been characterised by, amongst others, rapidly increasing global temperatures over the 20th and 21st centuries, i.e. with $\tau_w \approx 200$ years. If $\tau_w > \tau_{c,i}, \tau_{r,i}$ – in other words individual tipping elements evolve over a faster timescale, for example coral reefs and the Arctic winter sea ice (see Table 1) – the probabilities $\boldsymbol{D}$ will first settle on the stable steady state Eq. (10) parametrised by the forcing. On the other hand, for many other elements that evolve over multiple centuries or even millennia, for example WAIS and the Greenland

Ice Sheet, the dynamics will be dominated by the relatively fast global warming. For mixed systems therefore, especially over finite times, computational simulations are necessary to complement the analytic results.

### 2.4 Tipping elements of the Earth system

Armstrong McKay et al. (2022) comprehensively assessed 16 tipping, through their extensive multi-disciplinary literature review, providing estimates on, amongst others, the tipping threshold and timescales. In total the authors had reviewed 33

possible tipping elements, however in our work, due to the limited amount of TE-TE interaction data available (Wunderling et al., 2024), we focus only on nine of them – six global core tipping elements, two regional elements (low-latitude coral reefs



**Table 1.** Summary table of tipping elements ($TE_i$) and their respective parameters ($\tau_{c,i}, \tau_{r,i}, T_{lim,i}, \beta_i$) as inferred from the literature review of Armstrong McKay et al. (2022). Given the lack of knowledge regarding specific numerical values for the recovery timescales, we assume all recovery times are at least an order of magnitude larger than their respective collapse times, i.e. $\tau_{r,i} = 10\tau_{c,i} \; \forall i \in V$ – in doing so we also respect the (ir)reversibility of the tipping elements summarised by the Global Tipping Points Report 2023 (Lenton et al., 2023). *ENSO tipping threshold was estimated as the midpoint of the range given in Armstrong McKay et al. (2022).

|  | **Tipping Element** | **Collapse (years)** | **Recovery (years)** | $T_{lim,i}$ (°C) | $\beta_i$ (°C$^{-1}$) |
|---|---|---|---|---|---|
| **REEF** | Low-latitude coral reefs | 10 | 100 | 1.5 | 13.813510 |
| **ASI** | Arctic winter sea ice | 20 | 200 | 6.3 | 3.837086 |
| **PERM** | Boreal permafrost | 50 | 500 | 4 | 6.906755 |
| **AMOC** | Atlantic meridional overturning circulation | 50 | 500 | 4 | 2.656444 |
| **WAM** | Sahel & West African monsoon | 50 | 500 | 2.8 | 8.633443 |
| **AMAZ** | Amazon rainforest | 100 | 1,000 | 3.5 | 4.604503 |
| **ENSO** | El Niño-Southern Oscillation | 100 | 1,000 | 4.5* | 4.604503 |
| **WAIS** | West Antarctic ice sheet | 2,000 | 20,000 | 1.5 | 13.813510 |
| **GIS** | Greenland ice sheet | 10,000 | 100,000 | 1.5 | 9.866793 |

and the Sahel & West African monsoon) and the El Niño-Southern Oscillation (ENSO) whose status as a tipping element has been disputed (Collins et al., 2019; Lee et al., 2021). As our model is agnostic to the precise definition of a tipping element – for example as one containing hysteresis or irreversibility – and regardless of its classification, we include ENSO in our case study

since it has direct, strongly destabilising impacts on other subsystems, notably on the coral reefs and on the Amazon rainforest (Wunderling et al., 2024). Table 1 summarises for each tipping element the set of parameters inferred from the review, namely the timescales for collapse and recovery, the tipping threshold and the sensitivity. For specific inference methods, especially for the latter, see Appendix A1.

For the interactions between tipping elements we rely on a similarly extensive literature review by Wunderling et al. (2024),

in which the authors had assessed four properties of 19 pairwise interactions: response type, response strength, level of agreement within the literature and level of evidence. Although this review dealt in qualitative terms to describe the response strength, a core foundation of the review comes from the quantitative probability factors of the expert elicitation (Armstrong McKay et al., 2022). We use a similar methodology from an earlier work (Wunderling et al., 2021a) to, first, convert the qualitative terms back into probability factors, thus allowing us to include the new interactions considered.

A major challenge here is, where there is high uncertainty with respect to individual tipping elements (Armstrong McKay et al., 2022), particularly for the value of the tipping threshold, deep uncertainty is rife where interactions are concerned (Wunderling et al., 2024). That is, in some cases, there is not even general consensus as to whether a dyadic interaction is destabilising or stabilising. For example, there are multiple opposing mechanisms which allow the West Antarctic ice sheet to either strengthen (Pedro et al., 2018; Li et al., 2023) or weaken (Stouffer et al., 2007) AMOC, while other works suggest




the disintegration of WAIS can avoid (Sinet et al., 2023, 2024) or at least delay (Sadai et al., 2020) the subsequent collapse of AMOC. Therefore, in order to reflect the uncertainty both from the original expert elicitation (Kriegler et al., 2009) and the more recent review (Wunderling et al., 2024), we associate with each pairwise interaction a uniform random distribution whose range is the interval of plausible PF values (see Fig. 1 for illustration or Table A2 for details). Note, that since a uniform random distribution maximizes entropy or uncertainty given a constrained interval, no further assumptions other than the range

of plausible PF values are needed. Moreover note that in this work, we discard all 2 interactions with both unclear response type and unclear response strength, as there is no clear, reasonable way to quantify such interactions. Given the 17 dimensional space of parameters, we use a Monte Carlo ensemble of simulations using the Latin hypercube sampling method (McKay et al., 1979), which covers the parameter space more efficiently than i.i.d sampling.

## 3 Results

### 3.1 Theoretical solutions

#### 3.1.1 Equilibrium solutions

In this subsection, we will derive the system-wide stationary distribution of our tipping elements network and show that it is always stable. A source node $s \in V$, i.e. a node which has no in-neighbours, behaves precisely as an independent 2-state Markov chain with $p = p_{c,s}$ and $q = p_{r,s}$ as it is unaffected by any other nodes. For completion we include the update equation

for $D_s[t+1]$ below, however the explicit the time-evolution $D_s(t)$ and stationary distribution $D_{*,s}$ can be derived directly from Equations (3) and (4) by simple substitutions.

$$D_s(t) = D_{*,s}(1 - (1 - p_{c,s} - p_{r,s})^{t-1}) + (1 - p_{c,s} - p_{r,s})^t D_s(0) \tag{8}$$

$$D_{*,s} = \frac{p_{c,s}}{p_{c,s} + p_{r,s}} \tag{9}$$

Assume now there exists a stationary joint-distribution $\boldsymbol{D}_* = (D_{*,i} : i \in V)$ for the entire system which satisfies, for each

$i \in V$, $D_{*,i}[t+1] = D_{*,i}[t]$. Plugging this ansatz into the update equation in Eq. (6) results in $|V|$ simultaneous equations as seen in Eq. (10); although there is no explicit, closed-form expression the coupled equations can be solved numerically. Notice the solution for source nodes, $s \in V$, can also be derived from these equations directly by noting that all $a_{j,s}^{\pm} = 0$. Due to the logistic dependence of $p_{c,i}$ on global warming $\Delta T$, the equilibrium probability to be degraded $D_{*,i}$ is also dependent on global warming. In the limit of low temperatures, all $p_{c,i} \to 0^+$ such that the equilibrium Degraded risk similarly tends to 0.

As temperature increases, Eq. 10 increases monotonically until it converges to a high temperature limit $D_{*,i}^{hot}$ from below, since all $p_{c,i} \to (\tau_{c,i} + 1)^{-1}$ also from below.

$$D_{*,i} = \frac{(1 + \sum_{j \in V} a_{ji}^+ D_{*,j}) p_{c,i}}{(1 + \sum_{j \in V} a_{ji}^+ D_{*,j}) p_{c,i} + (1 + \sum_{j \in V} a_{ji}^- D_{*,j}) p_{r,i}} \tag{10}$$



### 3.1.2 Stability analysis

Having found a system-wide stationary distribution, we now derive conditions for its stability. A sufficient condition for the
stability of the entire network is to require each individual MC to be stable, that is $\forall i \in V$ to meet the condition that $0 < \tilde{p}_{c,i}|_* + p_{r,i} < 2$ where $\tilde{p}_{c,i}|_*$ is the effective collapse probability evaluated at $\boldsymbol{D}_*$. Notice that since the internal recovery probability is restricted to $p_{r,i} \in (0,1)$ the condition simplifies to $0 < \tilde{p}_{c,i}|_* < 1$. We will now find bounds for the sums $\sum_j a_{ji}^{\pm} D_j$ in order to simplify the stability condition further. In particular since $D_i \in [0,1] \,\forall i$, then $0 \le \sum_j a_{ji}^{\pm} D_j \le \sum_j a_{ji}^{\pm}$. Because of this we can find lower and upper bounds for the fraction in Eq. (5) as the following,

$$0 < \frac{1}{1+\sum_j a_{ji}^-} \le \frac{1+\sum_{j\in V} a_{ji}^+ D_{*,j}}{1+\sum_{j\in V} a_{ji}^- D_{*,j}} \le 1+\sum_j a_{ji}^+$$

where the first strict inequality is due to all probability factors being positively valued i.e. $a_{ij}^{\pm} \ge 0 \,\forall i,j \in V$, thus automatically fulfilling the first stability criterion, as the lower bound is strictly larger than 0.

Finally, since $p_{c,i}$ is monotonically non-decreasing in temperature, we can upper bound the collapse probability by the hot temperature limit, in terms of the collapse timescale and the length of a model time step, $p_{c,i} \le (1+\tau_{c,i}/\tau_m)^{-1}$, such that we
have the following inequalities.

$$\tilde{p}_{c,i}|_* \equiv \frac{1+\sum_{j\in V} a_{ji}^+ D_j^*}{1+\sum_{j\in V} a_{ji}^- D_j^*} p_{c,i} \le (1+\sum_j a_{ji}^+) p_{c,i} \le \frac{1+\sum_j a_{ji}^+}{1+\tau_{c,i}/\tau_m}$$

For the second criterion $\tilde{p}_{c,i}|_* < 1$ we can now compare the final expression above to 1 to arrive at the condition for $\boldsymbol{D}_*$ to be a stable stationary distribution.

$$\boldsymbol{D}_* \text{ stable for } \sum_j a_{ji}^+ < \frac{\tau_{c,i}}{\tau_m} \quad \forall i \in V \tag{11}$$

Although both probability factors $a_{ji}^+$ and collapse timescales $\tau_{c,i}$ are determined from the physical system, the model time step $\tau_m$ is a free parameter, independent of the real world system. The above stability condition can therefore be used to give an upper bound for the length of the model time steps such that the equilibrium solution Eq. (10) is always stable; in other words so long as the model time step is at most $\tau_m < \min_{i\in V}(\tau_{c,i}/\sum_j a_{ji}^+)$, we can guarantee the stationary distribution is stable. Note that with our parameters this minimum occurs for the coral reefs and by setting $\tau_m = 1$ year, our network of interacting
tipping points can almost surely (i.e. with probability 1) satisfy the stability condition.

### 3.1.3 Convergence times

Having satisfied the stability condition, we now consider the time taken for the system to converge towards the equilibrium solution. In particular let $t_\epsilon$ be the first time step at which the probability to be degraded $D(t)$ gets $\epsilon$ close to the equilibrium solution $D_*$, in other words $|D(t_\epsilon) - D_*| = \epsilon \ll 1$. We can approximate the evolution of the entire system by that of a single
source node with collapse time $\tau$ (evolving under Eq.(8)) and recovery time $10\tau$ – given our parametrisation assumptions



(see Table 1 for details) – where $\tau$ is the largest collapse time in the whole network. In so doing we can find an approximate expression (for the full derivation see Appendix C) for $t_\epsilon$ in terms of this maximal collapse time.

$$t_\epsilon \approx \frac{10}{11}\tau \ln \epsilon^{-1} + O(1) \tag{12}$$

For our network of tipping elements, for a threshold of $\epsilon = 10^{-3}$ this convergence time is approximately $t_\epsilon \approx 63\text{kyr}$. In other
words over millennial timescales, the long term behaviour of our system of 9 tipping points falls nears roughly 0.1% (in risk percentage) of the equilibrium solution. Thus in this work any mention of "long-term" refers to millennial timescales at least.

## 3.2 Equilibrium risks under fixed global warming levels

Far from being a neat theoretical result, the stable stationary distribution of Eq. (10) can give us valuable insights into interacting tipping elements in the Earth system. In particular, as the distribution is always stable, this distribution is precisely the long-term
risk, after millennial times, of each tipping element being Degraded. In this subsection, we explore this equilibrium or long-term behaviour for a range of fixed GMST anomalies in terms of the expected number of degraded tipping elements (Fig. 2) and the individual risk (Fig. 3). In the former, we see that the equilibrium response of tipping elements to the global warming level is highly non-linear, due in large part to the sigmoidal temperature-dependence of each element's internal collapse probability. If we were to permanently (i.e. at least over millennial timescales) commit to the Paris Agreement global warming levels of
$1.5 - 2°C$, then 3 tipping elements – both polar ice sheets and the low-latitude coral reefs – are very likely ($> 90\%$) to be in their respective degraded states as their individual critical thresholds have been surpassed.

Interactions between tipping elements on the whole increase the risk of being degraded, as indicated by the no-interactions case (dashed lines) generally lying below the ensemble mean for the with-interactions case. This is largely due to the majority of TE-TE interactions being destabilising (Wunderling et al., 2024); for some tipping elements, in fact, all of their known
incoming interactions are destabilising (see Table A2) such that interactions can only ever increase stationary risks. AMOC, although mostly dominated by its destabilising neighbours, has a moderately strong though of unclear type interaction due to WAIS, meaning that for some interaction matrices, the *sans* interaction scenario is riskier. Similarly, as AMAZ only has a single incident edge, also of unclear response, the interactions can decrease the risk; mostly, however, since its only neighbour is AMOC, the interactions typically increase the risk to be degraded.

In Fig. 3 we display the stationary risk-response for each individual tipping element. For all elements, when global warming is near pre-industrial levels ($\Delta T \leq 1°C$) the risk to be Degraded is zero or negligible. At very high temperatures ($\Delta T = 10°C$), the degraded risk for all tipping elements have more or less converged to the respective $D_{*,i}^{hot}$ discussed previously in Sect. 3.1. Most tipping elements under such catastrophically hot temperatures are very likely ($D_{*,i}^{hot} > 90\%$) to be degraded. Note that, as cab be seen in Eq. (10), $D_{*,i}^{hot}$ can never be precisely 100% unless $p_{r,i} = 0$ (or equivalently $\tau_{r,i} = \infty$) in other words unless the
tipping element is precisely irreversible. In other words even at high equilibrium temperatures – e.g. at $10°C$ global warming – a tipping element's equilibrium risk will not necessarily be 100% due to this property of the Markov chain model. Moreover for an individual tipping element $i$, having more destabilising neighbours increases $D_{*,i}^{hot}$ while if there are enough stabilising neighbours $D_{*,i}^{hot}$ will be reduced. We see this in the case of both the Greenland ice sheet and Arctic sea ice; the risk of both at



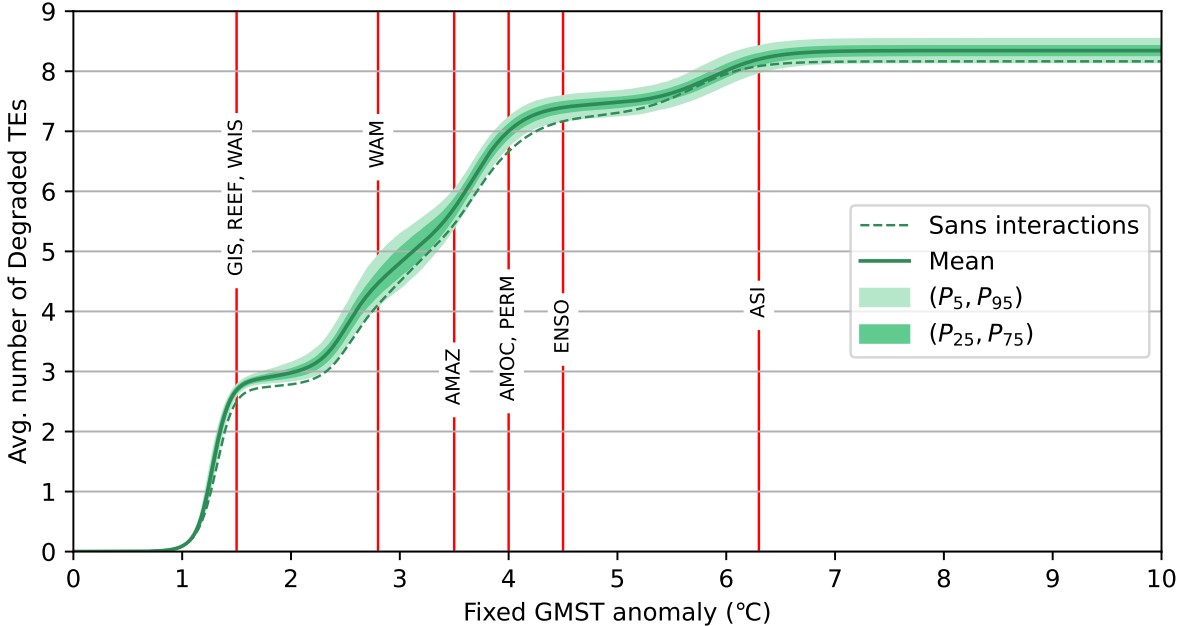

**Figure 2.** Expected number of degraded tipping elements at equilibrium, under a range of fixed levels of GMST anomaly. Specifically, for each value of fixed GMST anomaly, the expected number is simply the over all tipping element's degraded risk, with statistics done over the ensemble of 500 interaction matrices. The dashed line indicates the sans interactions case; bold lines the ensemble mean values; while the coloured bands shows values the 5- to 95-percentile uncertainty range in lighter hue, and the first- to third-quartile uncertainty range in darker hue; vertical red lines indicate the individual critical thresholds of different tipping elements.

very high temperatures is lower, though still likely ($D_{*,i}^{hot} \approx 80\%$), due to being partially stabilised by AMOC. We stress that,

assuming all thresholds have been crossed, increasing temperature monotonically increase the equilibrium risk.

As temperature increases the long-term (i.e. equilibrium) risks for all but one element increase monotonically, from which we see a threshold effect emerging. That is, below some global warming level – within the estimates of the critical threshold from Armstrong McKay et al. (2022), which is denoted by the dotted vertical lines in Fig. 3 –, an individual tipping element's risk nears $0$ while above this threshold the risk nears $D_{*,i}^{hot}$. The effects of interactions, on the other hand, are highly non-linear

in temperature, being most strongly felt over intermediate temperatures, especially around the critical thresholds of one or both elements involved. For example, as seen in Fig. S1, the equilibrium risk for AMOC increases by around 30% at around $3°C$, while the risk for the Arctic sea ice is reduced by around $20\%$ at $6°C$.

The slight differences between the emergent thresholds and the estimates of Armstrong McKay et al. (2022) is a consequence of the probabilistic model and, to various degrees, the interactions between tipping elements. For instance the destabilising

interactions felt by the AMOC reduces its effective critical threshold by about $0.4°C$. On the other hand, as the Arctic sea



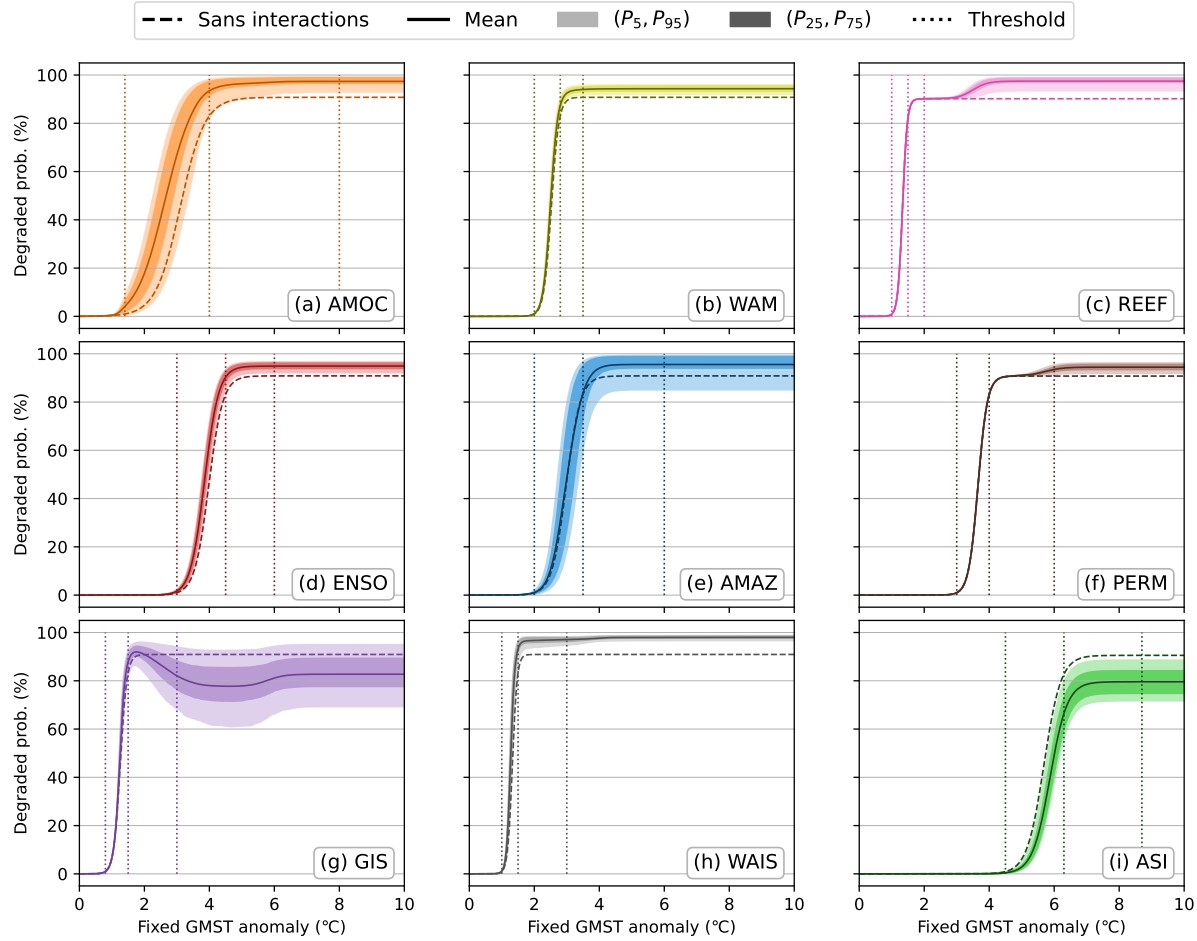

**Figure 3.** The stationary probability to be Degraded (see Eq. (10)) of each tipping element, for a range of fixed GMST anomaly values from $+0°C$ to $+10°C$. Dashed lines indicate the sans interactions case; bold lines the ensemble mean values; while the coloured bands showing values the 5- to 95-percentile uncertainty range in lighter hue, and the first- to third-quartile uncertainty range in darker hue. The thin dotted lines indicate the tipping threshold of each element (see Table 1).

ice has a single incoming interaction from the AMOC, which is moreover stabilising, its effective critical threshold is in fact increased, relative to no interactions, by around $0.2°C$.

The Greenland ice sheet stands out most amongst its counterparts as it shows a non-monotonic response to fixed global warming levels. Equilibrium risk increases with temperature until around $1.7°C$, after which the risk in fact decreases with
temperature in the range $1.7 - 4°C$. This corresponds to the range of temperatures over which the risk of AMOC quickly increases, thus helping to somewhat stabilise GIS. Between $4 - 5°C$ the risk plateaus at around $77\%$ as no other thresholds are being crossed, while between $5 - 6°C$ the Arctic sea ice crosses its critical threshold and destabilises GIS. Another consequence





of ASI crossing its threshold can then be seen in the corresponding rise in the permafrost risk to be degraded up to $D_{*,i}^{hot} \approx$ 97.5%.

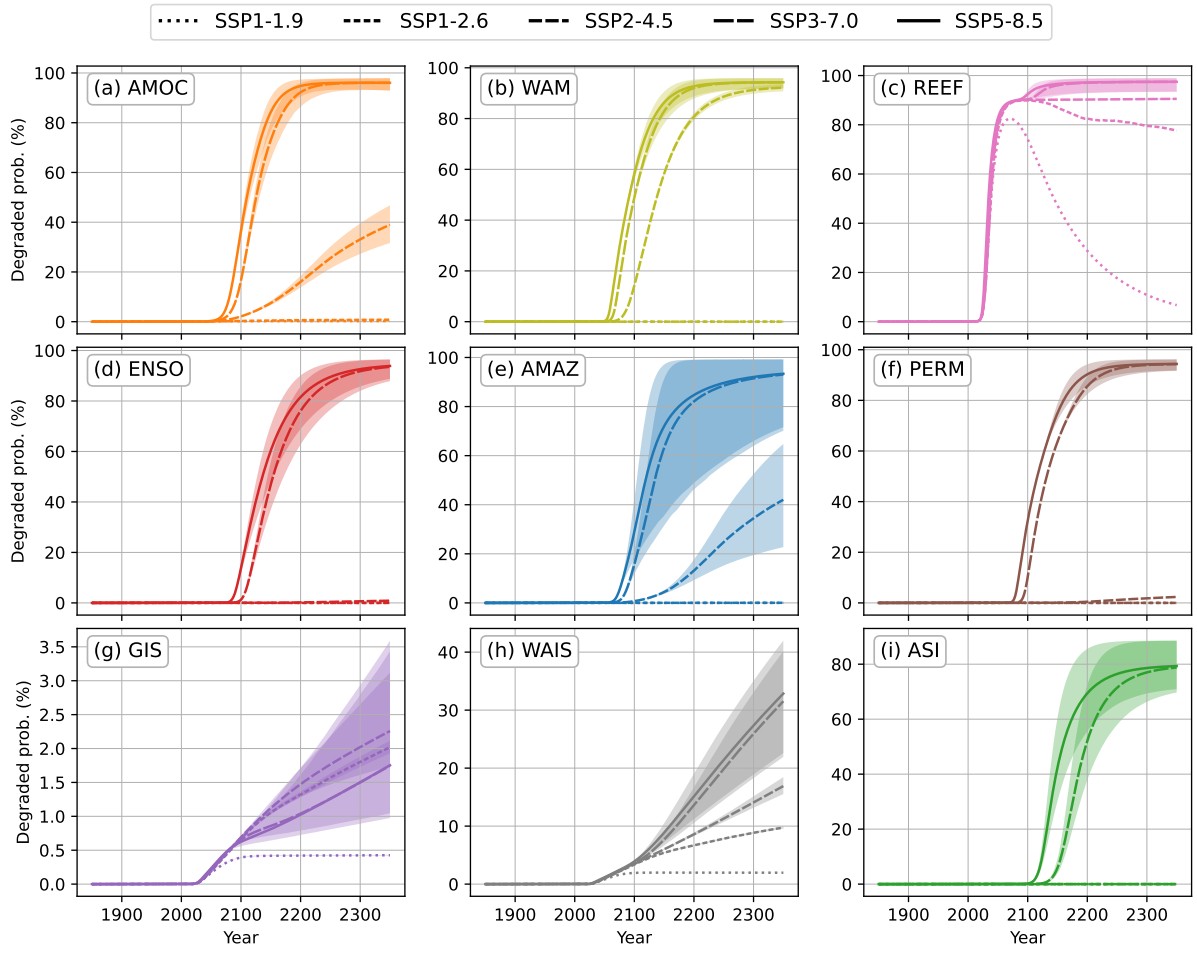

**Figure 4.** Ensemble mean of degraded probability of interacting tipping elements over time under different global warming scenarios, relative to 1850–1900 (Morice et al., 2021; Lee et al., 2021). In particular, for the years 1850–2014 historical warming is used while from 2015–2350 the projected global warming trajectories are used. For graphical clarity, only the 5- to 95-percentile uncertainty range of the ensemble are included, moreover comparisons to the *sans* interaction case has been left to Fig. S3.

## 3.3 Short term risks under the shared-socioeconomic pathways

While the stable stationary distribution highlights the long term risks for each tipping element, given that global warming level is fixed in time, the short term evolution is of most importance to policy and human life. For instance, it took the Earth less than a century to reach an annual GMST of +1.6°C in 2024 (Copernicus Climate Change Service, 2025) from around





+0.25°C in 1950 (Lee et al., 2021). In this section, therefore, using both historical levels of global warming (1850-2014) and
future projections (2015-2350) under the five extended shared-socioeconomic pathways (SSPs) outlined in the IPCC's Sixth
Assessment Report (Lee et al., 2021), we look at how the risk of each tipping element evolves over the course of 500 years
(from 1850 to 2350). As different SSPs represent different policy and commitment scenarios for the 21st century onwards, we
can see in Fig. 4, in particular, how different tipping elements respond, in essence, to human activity.

As the dynamics are dominated by the faster timescales of GMST evolution rather than the slower timescales of an individual
element's recovery or collapse, which SSP scenario the tipping elements are subjected to has a huge impact on the risk of being
Degraded certainly by 2350 and even in 2100 for some. For instance, under the fossil-fuelled scenario of SSP5-8.5, in which
exploitation of global resources are intensified and lifestyles become increasingly energy-intensive, the Sahel and West African
monsoon (WAM) has a 58% likelihood on average to be Degraded in 2100 and is very likely (94%) to be so in 2350. On the
other hand, under the sustainable and 'green' development pathway of SSP1-1.9, where ecologies are preserved, planetary
boundaries respected and income inequality being actively reduced, on average WAM is exceptionally unlikely ($< 0.003\%$)
to be degraded. Suffice to say, in all cases, following such a green and sustainable scenario minimises the risk of degrading
tipping elements, as compared to other scenarios.

That being said, under the most optimistic scenario SSP1-1.9, the coral reefs are at least likely to be degraded (>74%) in
2100 (see Fig. 5 due to the initial overshoot in temperature and their decadal collapse times. Fortunately, as the overshoot is
temporary, the coral reefs are able to recover sufficiently quickly that by 2350 (see Fig. S2) they are very unlikely ($< 10\%$)
to remain degraded. Although its singular interaction from ENSO does destabilise it over time, the increase from the sans
interaction case is only by less than 1% for the first three scenarios and by almost 7% for SSP3-7.0 and SSP5-8.5 (see Fig.
S3); the vast majority of its risk is due to its fast timescales (a few decades) and to its low tipping threshold (+1.5°C). On the
other hand for very slow tipping elements their dynamics are dominated instead by the evolution of SSP warming and their
faster neighbours. For instance, GIS, which evolves over millennia, is stabilised quickly enough by AMOC, which evolves
over decades to centuries, such that its 2100 risk is actually reduced by around 1% for the hotter scenarios. Intriguingly, under
both SSP1 scenarios, the ensemble mean risk is instead slightly increased, though by a very small amount, although some
interaction matrices still result in a reduction.

## 4 Conclusions

In summary, we find that global warming temperatures play a critical role in minimising the risk of tipping element degradation
in our novel risk assessment framework of nine interacting tipping elements across the Earth system. If we want to risk fewer
than 3 degraded tipping elements in the long term we must keep temperatures below 1.5°C – less than the lower limits of the
Paris Agreement – while in the short term we must commit to 'taking the green road' (Riahi et al., 2017) with regards to global
development and economic paradigms, to completely avoid any tipping risks. In particular, as exemplified by the SSP1-1.9
scenario (Lee et al., 2021) – one of the most optimistic overshoot scenarios – temporary overshoots in temperature mean that




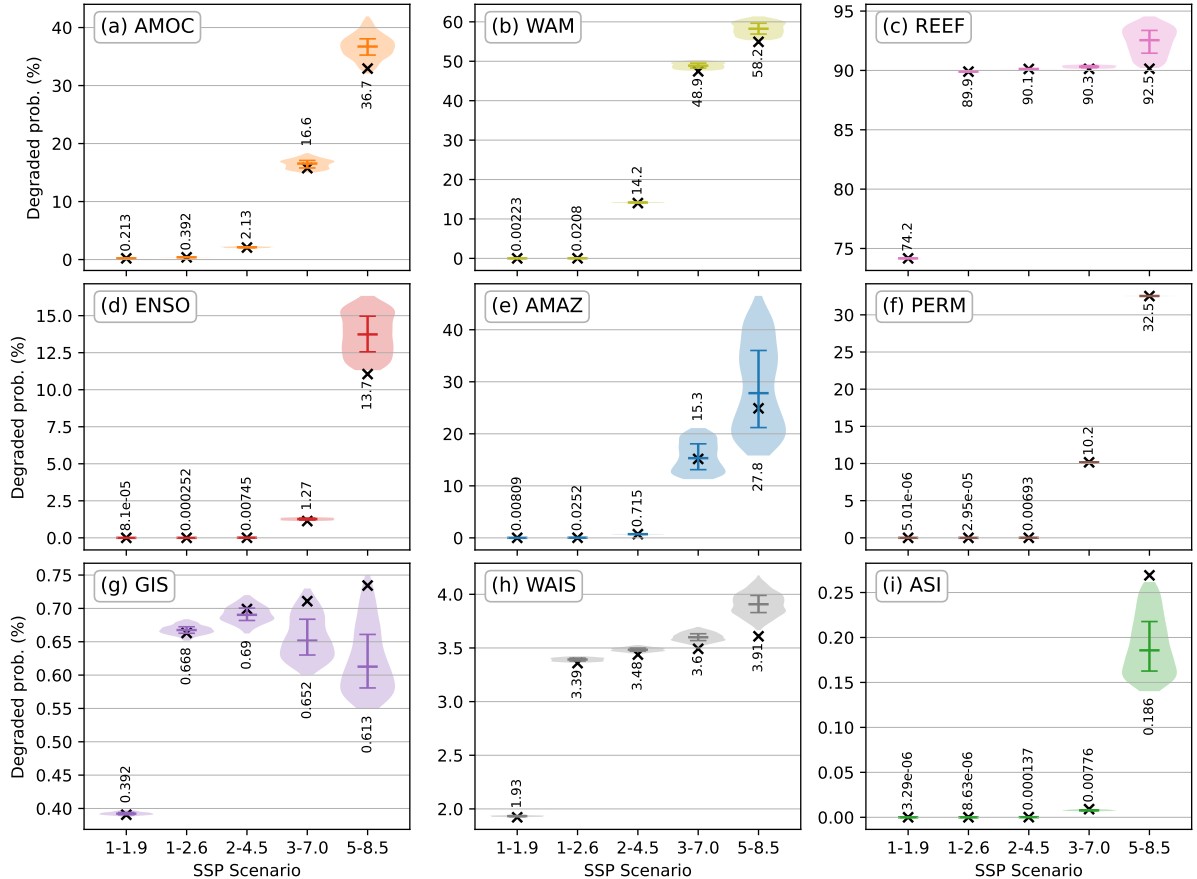

**Figure 5.** Distribution of Degraded risks of interacting tipping elements under different global warming scenarios by the year 2100. In particular, in each panel, each violin plot shows the distribution over the ensemble of plausible interaction matrices, with bars indicating the interquartile range and median, whose value is further indicated. Black crosses meanwhile indicate the sans interaction degraded risks, also evaluated at 2100.

the coral reefs will be very likely (94%) to be in their prosperous state by 2350, despite being, first, likely (74%) to be degraded in 2100.

Our framework differs from other works on interacting tipping elements (Gaucherel and Moron, 2017; Sinet et al., 2023, 2024; Wunderling et al., 2021a, 2023) as we model the interactions more coherently to the limited data, originally quantified from expert elicitation (Kriegler et al., 2009) as *probability factors*, while greatly expanding the set of elements to those found by Wunderling et al. (2024). In doing so, we find that interactions generally increase the risk of an element triggering, with two notable exceptions. The Greenland ice sheet and the Arctic winter sea ice are both stabilised by the AMOC such that their equilibrium risks are reduced by 10-20% in absolute value at equilibrium (see Fig. S1). Meanwhile in the short term, the latter,





under even the most high-emissions scenarios (SSP3-7.0 and SSP5-8.5), also undergoes a reduction in risk of around 25% in
the next century, due to this interaction (see Fig. S3).

By taking a probabilistic approach, we have significantly reduced the computational load while qualitatively mirroring many
of the results found by more complex models. At the macro-level, we find that, for the most part, interactions increase the risk
of an element tipping as other models have found (Wunderling et al., 2021a, 2023), while we also reproduce some tipping
element-specific results. For example, we find that the risk of the Greenland ice sheet tipping decreases over equilibrium
temperatures between $2 - 4°C$ while the equivalent risks for AMOC and the Amazon rainforest increase significantly within
this range as Wunderling et al. (2023) found for their system of only four tipping elements, as compared to our nine. For our
ensemble of 500 interaction matrices, numerically solving for the long-term risk of all tipping elements simultaneously across
$10^4$ values of temperature took 15 minutes on a personal computer. For the short-term risk over all SSP scenarios, given the
yearly resolution in time, all the simulations took less than 5 seconds.

While the Earth system as a whole is largely destabilised by interactions (Wunderling et al., 2024), some works have instead
focused on and highlighted the importance of sub-systems containing stabilising interactions and negative feedback loops. In
a conceptual model linking both polar ice sheets to the AMOC via saltwater flux, Sinet et al. (2023, 2024) found that AMOC
itself can effectively be stabilised depending on the collapse rates of and time delay between tipping of both ice sheets, even
without the cooling effects a weakening AMOC has on the Greenland ice sheets (Jackson et al., 2015). An early Boolean model
(Gaucherel and Moron, 2017), for instance, suggested that the negative feedback loop between the Atlantic meridional ocean
circulation and the Greenland ice sheet may be sufficient to stabilise the entire climate system, however this has been largely
refuted by more complex dynamical models (Wunderling et al., 2021a, 2023).

We find that the destabilising interactions dominate the stabilising effects of WAIS on AMOC. With regards to AMOC itself,
we find that on the one hand AMOC is still at worst about as likely as not (40%) to tip by 2100 under SSP5-8.5, despite the
inclusion of other, mostly destabilising, interactions – depending on the SSP scenario this risk may be even less than 5%. On
the other hand, by 2350, we can see how dependent the risk of tipping is on the specific SSP scenario, with the two hottest
scenarios (SSP3-7.0 and SSP5-8.5) leading to AMOC being very likely to tip, while under both SSP1 scenarios the risk is
negligible. We find that interactions on the whole play a destabilising role in AMOC though these effects are most prominent
only after 2100, increasing the risk by 10-15% on average and by up to 18% (see Fig. S3). Although under the other 3 scenarios,
some interaction matrices can reduce the tipping risk by up to 5% in 2350 – and thus somewhat stabilising AMOC as Sinet
et al. (2023, 2024) claimed – we find that the destabilising interactions not considered by Sinet et al. (2023, 2024) dominate
the stabilising effects of WAIS producing a net increase in risk by 2350 across all scenarios.

Although this subsystem cannot sufficiently stabilise itself nor the entire Earth system (see Fig. S4), individual tipping
elements are less at risk, in large part due to the stabilising effects *from* AMOC. For instance, for equilibrium temperatures
hotter than $2°C$, GIS is on average less likely to tip than without interactions. In the short term too, especially after 2100,
interactions lower the risk by up to 2%, depending on the scenario (see Fig. S3). Moreover due to its millennial timescales and
despite its lower threshold of $1.5°C$, GIS is very unlikely to tip (<3.5%) across all scenarios by 2350, thus allowing for some
overshoots in temperature as has been previously found (Ritchie et al., 2021; Wunderling et al., 2023). Similarly, the Arctic



sea ice too is stabilised greatly by the AMOC, both at equilibrium (by up to 20%) and in the short term (for the SSPs 3-7.0 and
5-8.5 the reduction is most prominent in the period 2150–2200 at around 20-30%).

Although large tipping risks can be avoided by limiting the convergence temperature – for long-term GMST under $1°C$, we
find all nine tipping elements are very unlikely to be tipped, similarly to Wunderling et al. (2023) – we are still well above such
temperatures, having reached $1.6°C$ in 2024 (Copernicus Climate Change Service, 2025). We are, however, already close to
having between 1 - 3 tipped elements – the coral reefs (>90%), the Sahel and West African monsoon (>48%) and the AMOC
(up to 36.7%) – by 2100 and by 2350 under the SSP3-7.0 and SSP5-8.5 scenarios this drastically increases to almost 7 tipped
elements (all except the Greenland ice sheet and perhaps the West Antarctic ice sheet). Even if global warming temperatures
are permanently maintained within the Paris Agreement limits of $1.5 - 2°C$, on average 3 elements (the coral reefs and the
polar ice sheets) will still be degraded, highlighting the fact that such limits must be treated only as short-term overshoot limits
which cannot be sustained in the long term.

While the coral reefs are at most risk both at equilibrium and in the short term, they form a sink node in the interaction
network – that is its only edges are incident – it affects no other nodes in the network, and thus any intervention or policy
directed at them would be limited in effect only to the reefs. A long-term and sustainable policy must instead take into account
and consider the entire network. For example, we find that AMOC has the highest betweenness centrality (Freeman, 1977),
regardless of the network instantiation, further supporting the findings of Wunderling et al. (2021a) that AMOC is a dominant
mediator of tipping cascades. Given limited resources, AMOC might therefore be an effective node to exogenously stabilise
or intervene on first. On the other hand, we stress, once again, the critical importance of limiting global warming and global
emissions of CO2 and other short-lived climate pollutants. Future work is needed, therefore, to rigorously test how interventions
on single nodes affect the short- and long-term dynamics of the entire Earth system network.

Increasingly, more research is being devoted to tipping points in the Earth system, of which our work is one. One major
work is the Global Tipping Points (GTP) Report 2023 (Lenton et al., 2023) that extensively focuses on all aspects and variety
of tipping points. Another international collaboration is the the Tipping Points Modelling Intercomparison Project (TIPMIP)
(Winkelmann et al., 2025) which seeks to systematically asses tipping risks using state-of-the-art coupled Earth system models,
following the success of the Coupled Model Intercomparison Projects (CMIPs) used for climate projections. As these large
projects continue to output detailed and complex models and simulations – the second iteration of the GTP Report is scheduled
to release during COP30 while some of the preliminary Tier 1 results and experimental protocols of TIPMIP are being released
– it will be important to compare the outputs of those models to our simplified framework, given the significant difference in
computational and time costs.

In this work, we have used the SSP warming scenarios as proxies for the potential effects of humans on the environment and
ecology. However, given how complex social-environmental relationships are, it is necessary to integrate agency more directly.
Eco-evolutionary games (Tilman et al., 2020), multi-agent reinforcement learning dynamics in stochastic games (Barfuss et al.,
2019, 2020, 2025), and networked and spatial games (Bara et al., 2024) are promising frameworks to do so. Our networked
model lays the groundwork for future works that consider the social dilemma of interacting Earth tipping elements.



**Appendix A: Data methods**

**A1 Parameter inference from threshold and timescale estimates**

Our individual tipping element data is based on the work of Armstrong McKay et al. (2022), who had reviewed 33 tipping elements. However in our work, due to the limited amount of TE interaction data available (Wunderling et al., 2024), we consider only nine of the 33 tipping elements – 6 global core tipping elements, 2 regional elements (low-lattitude coral reefs and the Sahel & West African monsoon) and El Niño-Southern Oscillation (ENSO) whose status as a tipping element has been disputed (Collins et al., 2019; Lee et al., 2021).

In some cases model parameters can be directly taken from the original dataset; the collapse timescale $\tau_{c,i}$ is precisely the "Timescale (y) - Est." while the midpoint of the logistic response function $T_{lim,i}$ is given by the "Threshold (°C) - Est.". In the event there is no estimate, for either parameter, then the average of the upper and lower bounds can be used instead – for example although there is no threshold estimate for ENSO, it has a minimum temperature of 3.0°C and 6.0°C, hence we estimate $T_{lim,i} = 4.5$°C.

Given climate model limitations and the historical rarity of a tipping point collapsing into a degraded state – let alone recovering from one back into the prosperous state – very little data, almost none, exists regarding the recovery timescales. Instead of numerical values for specific recovery timescales, the Global Tipping Points Report 2023 (Lenton et al., 2023) instead considered whether a tipping element was 'irreversible', at least over decadal or centennial times. For instance the polar ice sheets (with high confidence) and the Amazon rainforest (with medium confidence) are considered 'irreversible',

while the coral reefs are thought to only be irreversible (with medium confidence) over decadal timescales. On the other hand the Arctic winter sea ice is considered reversible (with medium confidence) even at these shorter timescales. In this work, therefore, we assume an order of magnitude difference between the collapse timescales and the recovery timescales which are sufficient to meet the (ir)reversibility conditions summarised in the Global Tipping Points Report (see Table A1).

Finally for the logistic growth rate parameters $\beta_i$ of each tipping element we utilise the literature derived bounds for the
threshold temperature. For all considered tipping elements, lower bounds for the threshold estimate exist, and as such we we fix $\beta_i$ such that the collapse response function is $0.001$ at the lower bound and $0.5$ at the threshold estimate. Mathematically this reduces to, for $\sigma(T_{min,i}; \beta_i) = \sigma_{min}$, such that we can solve for $\beta_i = \log(\sigma_{min}^{-1} - 1)/(T_{lim,i} - T_{min,i}) = \log(999)/(T_{lim,i} - T_{min,i})$.

**A2 Probability factors between tipping elements**

While the individual data of tipping elements are largely quantitive the extended data set for TE-TE interactions of Wunderling et al. (2024) are qualitative. In particular for each pairwise TE-TE interaction (i.e. edge), Wunderling et al. (2024) had summarised their extensive literature review into four qualitative variables (for details see their Table 1 and Figure 3): **Response** (Destabilising, Stabilising or Unclear); **Response strength** (Strong, Moderate, Weak or Unclear); **Agreement within literature** (High, Medium or Low); **Evidence** (Robust, Medium, Limited or Very Limited).





**Table A1.** (Ir)reversibility of individual tipping elements as reproduced from the Global Tipping Points Report 2023 (Lenton et al., 2023).

| Tipping Element | Irreversible? (decadal / centennial) | Confidence level | Source table in Lenton et al. (2023) |
|:---:|:---:|:---:|:---:|
| REEF | Yes (decadal) | Medium | Table 1.3.1 |
| ASI | No | Medium | Table 1.2.1 |
| PERM | Yes (wrt to carbon loss) No (wrt to frozen soil) | High | Table 1.2.1 |
| AMOC | Yes (centuries) | Medium | Table 1.4.1 |
| WAM | Decades to centuries | Unclear | Table 1.4.1 |
| AMAZ | Yes | Medium | Table 1.3.1 |
| ENSO | Not a tipping element | Medium | - |
| WAIS | Yes | High | Table 1.2.1 |
| GIS | Yes | High | Table 1.2.1 |

Much of their interactions, however, are based on the original, quantitative expert elicitations (Kriegler et al., 2009), from which, in an earlier work, Wunderling et al. (2021a) had converted into edge weights for their dynamical systems model ((for details see their Table 2). Comparing these values to the qualitative interactions of the later work (Wunderling et al., 2024) allows us to reverse engineer a mapping from the qualitative interactions into probability factors, thus effectively extending the set of probability factors from Kriegler et al. (2009). Roughly speaking, the response type determines the sign of the edge,

i.e. whether the edge would contribute to either the numerator or denominator in Eq. (5), while the response strength affected the maximum possible weight of the edge. We clean the dataset slightly by discarding edges whose type and strength are both unclear, thus removing the (ENSO, AMOC) and (AMOC, Indian summer monsoon) edges. In so doing this also entirely disconnects the Indian summer monsoon from the rest of interaction network, as it has no other edges and will therefore neither be affected by or affect the other tipping elements, at least according to the review (Wunderling et al., 2024).

For each TE-TE interaction we randomly sample a weight $w_{ij} \sim U(m_{ij}, M_{ij})$ whose distribution is parametrised by a minimum $m_{ij}$ and maximum $M_{ij}$; the maximum *absolute* value is given by the response strength – "Strong" = 1; "Moderate" = 0.5; "Weak, Moderate" = 0.3; "Weak" = 0.2; "Unclear" = 0.15 – while the sign of the weight is given by the response type – "Destabilising" is positive, "Stabilising" is negative, "Unclear" can be either. Similarly edges with a clear type will have a minimum absolute value of with sign given by the edge type; unclear edges have a minimum absolute value equal to

the negative of the maximum absolute value. Table A2 summarises the distributions of $w_{ij}$ using the data from Table 1 and Figure 3 of Wunderling et al. (2024). Finally, with respect to our signed networks, edge $(i,j)$ is destabilising (stabilising) if $w_{ij} > 0$ ($w_{ij} < 0$), with edge weight $|w_{ij}|$. In this way TE-TE interactions with unclear responses can either have stabilising or destabilising edges, depending on the sampling.





**Table A2.** Table of TE-TE interactions including the minimum and maximum possible values for their edge weights, inferred from Table 1 and Figure 3 of Wunderling et al. (2024). Specifically, each interaction has weight $w_{ij}$ which is uniformly distributed from a minimum weight $m_{ij}$ to a maximum weight $M_{ij}$, in other words $w_{ij} \sim U(m_{ij}, M_{ij})$. The sign of the weight corresponds to the sign of the corresponding signed edge ($a_{ij}^{\pm}$) while the magnitude of the weight is the value of the signed edge, in other words $a_{ij}^{\text{sgn}(w_{ij})} = |w_{ij}|$.

| Source | Target | Response | Strength | $m_{ij}$ | $M_{ij}$ |
|--------|--------|----------|----------|----------|----------|
| GIS | AMOC | Destabilising | Strong | +0.1 | +10 |
| WAIS | AMOC | Unclear | Weak, Moderate | -3 | +3 |
| AMOC | GIS | Stabilising | Strong | -10 | -0.1 |
| AMOC | WAIS | Destabilising | Unclear | +0.1 | +1.5 |
| GIS | WAIS | Destabilising | Moderate | +0.1 | +5 |
| WAIS | GIS | Destabilising | Weak | +0.1 | +2 |
| ASI | AMOC | Destabilising | Weak, Moderate | +0.1 | +3 |
| AMOC | ASI | Stabilising | Weak, Moderate | -3 | -0.1 |
| ASI | GIS | Destabilising | Weak | +0.1 | +2 |
| ASI | PERM | Destabilising | Weak | +0.1 | +2 |
| AMOC | AMAZ | Unclear | Moderate | -5 | +5 |
| AMOC | ENSO | Destabilising | Weak | +0.1 | +2 |
| ENSO | AMAZ | Destabilising | Strong | +0.1 | +10 |
| ENSO | WAIS | Destabilising | Weak, Moderate | +0.1 | +3 |
| ENSO | REEF | Destabilising | Strong | +0.1 | +10 |
| AMOC | WAM | Destabilising | Unclear | +0.1 | +1.5 |
| PERM | AMOC | Destabilising | Unclear | +0.1 | +1.5 |

## A3 Warming under Shared Socioeconomic Pathways

The Shared Socioeconomic Pathways (SSPs) are a variety of potential scenarios for climate change under different global socioeconomic changes. In particular the five used in this work were produced in the IPCC Sixth Assessment Report (AR6) (Lee et al., 2021; Fyfe et al., 2021; Intergovernmental Panel on Climate Change (IPCC), 2023) and represent different large-scale narratives in terms of sustained global efforts to counter climate change. In particular, similarly to Armstrong McKay et al. (2022), we use the mean global surface temperature anomaly, relative to 1850-1900, though we use the extended scenarios,

with historical data (1950-2014) and the projections (2015-2350) under 5 different SSPs. For the sake of clarity, we have displayed only simulation results that evolve under the mean global surface air temperature (GSAT) under the SSPs.





## Appendix B: Computational methods

### B1 Interaction matrices

Given the uncertainty regarding the 19 different TE-TE interactions, we utilise a 19-dimensional Latin hypercube sample
(McKay et al., 1979) in order to give a better coverage of the 19-dimensional parameter space, using the LatinHypercube class
of the SCIPY Python package (Virtanen et al., 2020). In particular we take 500 samples using this method, each one representing
a different instance of a $9 \times 9$ interaction adjacency matrix. For consistency and comparison we use the exact same ensemble
in both the short-term and long-term risk analysis. Any ensemble statistic – e.g. mean, quantiles, etc. – are based on these
500 interaction matrices. In contrast, the sans interaction treatment needed no ensemble as, in effect, the adjacency matrix of
independently evolving tipping elements is the zero matrix $\mathbf{0}$.

### B2 Long- and short-term risk simulations

For the nine tipping elements we consider, Eq. (10) gives 9 simultaneous equations that we solve numerically using the fsolve
function of NUMPY Python package to derive the joint stationary distributions (i.e. the long term risk) for a range of global
warming temperatures in $[0, 10]°$C with incremental steps of $0.001°$C, for each of the 500 interaction matrices of the ensemble,
as well as for the sans interaction case.

In the short-term case, we simulate using the update equation in Eq. (6), the evolution of each tipping element annually given
each of the 500 interaction matrices and the 5 extended SSPs. For the years 1850 - 2014 (inclusive) the historical levels of global
warming were used, while all tipping element risks were initialised to $D_i(1850) = 0, \forall i$. For the years 2015 - 2250 (inclusive),
the 5 SSP scenarios continue from the historical scenario both in temperature and the individual risk of each element.
Since degraded-risks are bounded ($D \in [0, 100]\%$), the underlying distribution of such risks cannot be Normal- nor t-
distributed. As such we do not report confidence intervals per se – that is the interval above and below the ensemble (sample)
mean in which 90% of the time the true population mean would be captured, assuming the samples are Normal- or t-distributed
– since these may produce intervals which are not strictly within the bounds. Instead we report ranges within certain percentiles
of the ensemble, such that the 5- to 95-percentile range, for example, covers the middle 90% of ensemble simulations.

## Appendix C: Derivation of Eq. (12)

Consider a 2-state Markov chain with probability to collapse $p$ and probability to recover is $q$. Its probability to be degraded
$D(t)$ evolves via Eq. (3) and has equilibrium solution $D_*$ given by Eq. (4). Let $t_\epsilon$ be the convergence time to the equilibrium
solution, in other words the first time step at which $t_\epsilon = \min\{t : |D(t) - D_*| < \epsilon\}$. Assuming $p + q < 1$, $D(t)$ monotonically
increases with time $t$ towards $D_*$ so that we can approximate $t_\epsilon$ as occurring once $|D(t) - D_*| = \epsilon$. Given our parametrisation
assumptions from the real data (that the recovery time is ten times the collapse time) we can simplify $p = 1/(\tau + 1)$ and



$q = 1/(10\tau + 1)$.

$$|D(t_\epsilon) - D_*| = \left(1 - \frac{1}{\tau + 1} - \frac{1}{10\tau + 1}\right)^{t_\epsilon - 1} = \epsilon$$

$$\ln \epsilon = (t_\epsilon - 1) \ln \left(1 - \frac{1}{\tau + 1} - \frac{1}{10\tau + 1}\right)$$

$$\ln \epsilon = (t_\epsilon - 1) \ln \left(1 - \frac{11\tau + 2}{(10\tau + 1)(\tau + 1)}\right)$$

$$t_\epsilon - 1 = \frac{\ln \epsilon}{\ln(1 - x)}$$

where $x = (11\tau + 2)/((10\tau + 1)(\tau + 1)) < 1$ for $\tau > 1$. Since $x < 1$ we can use the following series expansion for $1/\ln(1 - x)$,

$$\frac{1}{\ln(1 - x)} \approx -\frac{1}{x} + \frac{1}{2} + \frac{x}{12} + O(x^2)$$

$$= -\frac{10\tau^2 + 11\tau + 1}{11\tau + 2} + \frac{1}{2} + \frac{1}{12}\left(\frac{1}{10\tau + 1} + \frac{1}{\tau + 1}\right) + O\left(\frac{1}{\tau^2}\right)$$

$$\approx -\left(\frac{10\tau}{11} + \frac{101}{121} - \frac{1}{11\tau}\right) + \frac{1}{2} + \frac{1}{12}\frac{11}{10\tau} + O\left(\frac{1}{\tau^2}\right)$$

$$= -\frac{10\tau}{11} - \frac{81}{242} + \frac{241}{1320\tau} + O\left(\frac{1}{\tau^2}\right)$$

where the first line can be derived either through a Laurent expansion or through a Taylor expansion first of $\ln(1 - x) = -\sum_{n=1}^{\infty}(x^n/n)$ followed by a Binomial expansion. This lets us approximate $t_\epsilon$ in terms of the collapse time $\tau$.

$$t_\epsilon \approx 1 + \left(\frac{10}{11}\tau + \frac{81}{242} - \frac{241}{1320\tau}\right)\ln\left(\frac{1}{\epsilon}\right)$$

*Code availability.* Code will be made available upon request.

*Author contributions.* JB and WB developed the model, JB derived the theoretical results, conducted model simulations and prepared the figures. All authors discussed the results, wrote and edited the paper.

*Competing interests.* The authors declare that they have no conflict of interest.

*Acknowledgements.* JB and WB acknowledge the support of the Cooperative AI Foundation. N.W. acknowledges the Center for Critical Computational Studies at Goethe University Frankfurt am Main and the Senckenberg Research Institute for providing funding for this research. N.W. is also grateful for support from the KTS (Klaus Tschira Stiftung) under the project DETECT (ID 25545).



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
