# Peer review of "A risk assessment framework for interacting tipping elements"

_EGUsphere, 2025_

## Referee Comment (RC1)

**Review report for "A risk assessment framework for interacting tipping elements"**

September 10, 2025

This manuscript introduces a fully probabilistic model to represent interactions among climate tipping elements and ENSO. Each subsystem is described as a Markov chain, and pairwise interactions are represented via a network, with strengths primarily parameterized from expert assessments (notably Kriegler, 2009). The authors' aim is to provide a coherent and computationally efficient framework for risk assessment, integrating parameterized interactions into long-term and scenario-based analyses.

The manuscript includes analytical results for equilibrium risks and extends these with simulations under SSP scenarios to draw conclusions for the next 300 years. The probabilistic approach to parameterized interactions constitutes a novel contribution, and the framework is an original and relevant contribution, with the potential to advance the literature on interacting tipping elements. However, in its current form, the manuscript has several significant weaknesses: the unclear treatment of ENSO, an insufficiently explained role of timescales, a barely present reflection on limitations, and issues with clarity and presentation. I am inclined to support publication, provided these substantial issues are addressed. See below for more details.

**Major Comments**

**M1: ENSO as a tipping element.** The study considers nine climate subsystems, including ENSO. While the inclusion of ENSO can be justified by its central role in the climate system, it is not generally recognized as a tipping element, and its classification remains disputed, as acknowledged by the authors. The manuscript, however, repeatedly refers to ENSO explicitly and implicitly as a tipping element (e.g., "the nine tipping elements"), in the text, tables, and figures. I understand that the difference is hard to make throughout the study but, as it is, this is not correct.

**M2: Timescales of tipping.** The role and interpretation of tipping timescales are not sufficiently clear, at least for me. As written, it is difficult for readers unfamiliar with this methodology to grasp how these timescales should be understood, and what they influence. Do they represent the time for a subsystem to transition between states? If so, this should be made explicit.

**M3: Conclusions and limitations.** The conclusion is lengthy and could be made more concise and focused. In my opinion, several parts belong in the results section. More importantly, the discussion lacks sufficient reflection on limitations. Key questions remain:

- Is a fully probabilistic representation appropriate for physical systems? Is it more suitable than, say, Wunderling 2021?
- I understand that the interaction is still linear between the subsystems. This is a limitation, similar to previous work, and has to be acknowledged as such.
- What are the implications of limited data quality?
- What new insights are provided beyond previous studies such as Wunderling 2021?

- At present, many of the conclusions appear to be direct consequences of the assumptions rather than emergent results (e.g., interactions destabilise the system, when it is hypothesised that most are destabilising). It is crucial to articulate clearly what the results obtained via this framework add to current knowledge.
- How far could one go in studying coupled tipping elements via this framework?

**M4: Readability.** The manuscript contains many convoluted sentences and vague formulations, which make the first read unnecessarily difficult for a paper that is otherwise not highly technical. I strongly recommend careful editing for clarity and precision. I have listed specific examples below.

**Specific Comments**

**Abstract**

1: Capitalize "Atlantic Meridional Overturning Circulation".
3: "collapse" and "into a degraded state" sound redundant; collapse alone should be enough.
5: Not sure, but isn't "literature-based expert assessment" more correct?
6: "equilibrium risks of tipping elements" is not understandable at this point.
6: "their stationary distribution" alone, referring to tipping elements, is vague.
8: Capitalize "Shared Socio-economic Pathway".
8: "Hypothetical" is not correct; I guess you mean "idealised".
11: More correct would be "SSP scenarios with the strongest greenhouse gas emissions".
12: What do you mean here by comprehensive? It does not seem fair to me. At least, in climate science, it is a confusing denomination.

**Introduction**

15: "Earth systems" is not correct. I think you mean "climate subsystems".
15–18: The second sentence is not really an example of the first. The initiation of the marine ice sheet instability beyond some tipping point is indeed a nonlinear effect, but does not have to do directly with either path dependency or chaotic behaviour.
21: Please explain what you mean by "levels".
25: More correct would be "most interactions between tipping elements are destabilizing".
26: Say at least "tipping cascades". I would say that here can be a moment to explain what a tipping cascade is.
27: Replace polar ice caps by "the WAIS". I understand that the sentence does not work with this change, but the paper does not say that ice caps stabilize the AMOC.
28: Here and in other places, "saltwater" should be "freshwater", or maybe "meltwater" if originating from an ice sheet.
33: Replace point by ,
35: I don't understand "system model".
31–44: Personally, I find that these two paragraphs are a bit of an overkill in this introduction. Or at least, they give me the feeling of taking too much space.
55–58: There is excessive use here of the —, this part is not readable. Please separate it into sentences, as it is I do not really understand what the authors are trying to say.
55: "noise-induces tipping in some climate models occur when" is not grammatically correct.
56: The description of N-tipping quoted from Ashwin 2021 introduces unnecessary technicality.
57: Maybe it is because of the weird phrasing, but I don't understand "or through confidence intervals on parameter values".
64: You do not remedy the shortcoming of linear interaction though, which is a big one when it comes to interacting climate tipping elements. As far as I understand, the interaction impacts the probability linearly.
77: "which accounts for the physical sensitivity of the element, or at least uncertainty in the critical temperature" is a very vague formulation, especially the "at least".
82: "including under external forcing" is awkward in this sentence.

83: see M1.

85: "equilibrium degraded probability" sounds like the probability is degraded. I imagine there must be a better formulation.

Figure 1: In the caption, see M1.

**Methods**

92: I am not very familiar with these methods, so here it could be useful to have one or two short sentences that describe what will be done. Like "We describe each tipping element by . . . ".

95: Here, I understand "sufficiently" should be "fully".

In General: it is more common for me to write equations as soon as they are mentioned.

102: I don't understand the notion of stability of an equation in this context. Is it the stability of the stationary distribution?

102: Absolute value of the Jacobian can mean a lot of different things, so it should be clear here.

109: "the adjacency matrix".

133: see M1.

135: It is not explained what the two timescales are. In regard to M2, it would be a good moment to explain why the timescales are involved at this moment, in this way, and why in the value of the internal probabilities. As I read it, I understand that probabilities evolve with respect to these timescales, which is not what a tipping timescale is (i.e. the time for transitioning from a state to another)

144: Here, it is not indicated how you set these parameters beta. I understand it is in an Appendix; refer to it.

148: Eq 10 appears many pages after; for me this is awkward.

Table 1: For the table and caption, see M1. Also, regarding the *, I did not manage to find this information in Armstrong. Finally, the index i should be explained.

161: see M1.

167: I imagine you refer here not to Armstrong but Kriegler?

170: Please reformulate the first sentence.

171: I would not use "rife", as it is not very non-English friendly. Same for dyadic, could use bilateral, or bidirectional.

180: What do you mean all 2 interactions?

180–181: Do you reflect on that later? It is a good reason to justify that you cannot include it, not that it should not be.

**Results**

188: I understand that the stability is rather the object of the next subsection.

238: I did not catch why eq 10 is "far from a theoretical result"; it seems to me that it is precisely what it is. Also, "into interacting tipping elements in the Earth system" could already be more specific.

247–248: Seeing this figure for the first time, I would judge that this variation is fairly weak on Fig 2, given the fact that most interactions are destabilising. I would like the authors to reflect on that.

252: Why the use of the French word "sans", also in the figure? I think "scenario without interaction" or "interaction-free scenario" would work.

Fig 2: Again, degraded risk sounds like it reads as if the risk is degraded. For the figure and caption, also see M1.

259: typo for "cab" → "can".

Fig 3: For the caption, see M1.

Fig 4: For the caption, see M1.

In section 3.3: these are "extended SSP scenarios", not "SSP scenarios".

**Conclusion**

315: This sounds like a direct consequence of the assumptions. Maybe you mean that, considering a robust representation of interactions, global warming remains the most important factor?

316: see M1.

332: What do you mean here by more complex models? In this paragraph, only Wunderling 2021 and

2023 are cited. Do you compare to these references? If yes, "more complex" is misleading.

332: In regard to M3, it sounds trivial, as most interactions are assumed to be destabilising.

342: saltwater should be freshwater or meltwater.

347: Here again, not sure how Wunderling is more complex than the presented model.

358: What does "nor the entire earth system" mean here? There is no representation of the Earth system in this model.

363: I guess the author means 'safe overshoot' here.

367: see M1.

395: I would begin with "to conclude", or similar.

**Appendices**

Around 435: It seems to me that the method for choosing interaction strength is very subjective, and is not really announced as such.

In general: I have seen only references to appendices A1 and C. Please refer to appendices when relevant.

**Suggestions**

3: I would write "In doing so, ...".

4: I would write "therefore, in this work ...".

9: Here I would write "... . For instance, ".

23: Maybe "each other".

24: I would change ripple for propagate.

150: I would write "therefore, for mixed terms, ...".

162: It would be clearer as "sensitivity to ...".

163: I would write "for the sensitivity".

166: I would rather write "although this review dealt with describing response strengths in qualitative terms ...".

180: I would write "Moreover, note that, in this work, ...".

---

## Author Comment (AC1)

**A risk assessment framework for interacting tipping elements**

Jacques Bara[1,2], Nico Wunderling[3,4,5], and Wolfram Barfuss[1,2,4,6]

[1]Center for Development Research, University of Bonn, 53113 Bonn, Germany
[2]Transdisciplinary Research Area Sustainable Futures, University of Bonn, 53115 Bonn, Germany
[3]Center for Critical Computational Studies, Goethe-University Frankfurt, 60322 Frankfurt am Main, Germany
[4]Earth Resilience Science Unit, Potsdam Institute for Climate Impact Research (PIK), Member of the Leibniz Association, 14412 Potsdam, Germany
[5]Senckenberg Research Institute and Natural History Museum, Member of the Leibniz Association, 60325 Frankfurt am Main, Germany
[6]Institute for Food & Resource Economics, University of Bonn, 53115 Bonn, Germany

**Correspondence:** Jacques Bara (jbara@uni-bonn.de) and Wolfram Barfuss (wbarfuss@uni-bonn.de)

First, we would like to start by thanking the reviewers for their thorough remarks, reviews and constructive suggestions. We also thank the editor for their time and effort. As an overview of what we will include in our revision, we will first improved readability, particularly in the introduction (removing two extraneous paragraphs) and in the conclusions (moving several paragraphs to the Results section and reordering the conclusions to have a smoother flow). Moreover we will add to the Conclusions the discussion points requested by both reviewers, namely: the limitations of our model, the suitability of a probabilistic approach towards physical systems, the limitations of uncertainty regarding intra-climate subsystem interactions and the effects due to the temperature independence of recovery.

In our Methods section, we will add clearer signposting at the beginning of the section and add a more in-depth description and discussion regarding the relationship between timescales and probabilities, adding extra emphasis on the non-linearity of stabilising interactions. In the Results section, we will move a few paragraphs that had been in the Conclusion into this section as well as move a supplementary figure (previously Fig. S3) into the main text (as future Fig. 6) alongside an extra paragraph describing the results specific to it. Finally we will clarify explicitly throughout the paper that ENSO is not a tipping point, instead referring to it as a climate subsystem to avoid any misunderstandings.

Below, we provide a point-by-point response, where we include our responses in blue to contrast with each comment by the reviewers in black. We are thankful for the careful read of our paper and the very thorough feedback and comments that helped us improve our contribution. We hope that our revisions fully address the reviewers' concerns and meet the high bar of quality of *Earth System Dynamics*.

**1 Reviewer 1**

**1.1 Major Comments**

**1.1.1 M1: ENSO as a tipping element.**

The study considers nine climate subsystems, including ENSO. While the inclusion of ENSO can be justified by its central role in the climate system, it is not generally recognized as a tipping element, and its classification remains disputed, as acknowledged by the authors. The manuscript, however, repeatedly refers to ENSO explicitly and implicitly as a tipping element (e.g., "the nine tipping elements"), in the text, tables, and figures. I understand that the difference is hard to make throughout the study but, as it is, this is not correct.

Thank you for this comment. We wish to first reiterate that indeed ENSO is generally not recognised as a tipping element and also do not wish to claim that it is. We had often used phrases like "all nine tipping elements" more for brevity than any claim or statement regarding ENSO's status. We will rectify this by replacing all such statements in the text, tables and figures with one of the following, or variants thereof: "all nine climate subsystems", "eight tipping elements and ENSO", etc. Moreover we will emphasise that modelling a climate subsystem as a Markov chain is not restricted to tipping elements at the beginning of Sect. 2, as well as how to interpret a collapse timescale for ENSO not per-se as a *tipping* timescale like the other subsystems, but instead as the transition time towards e.g. a state of increased El Nino amplitude and/or increased rainfall amplitude variability at the beginning of Sect. 2.3. We expand upon this further in our response to M2 below.

**1.1.2 M2: Timescales of tipping.**

The role and interpretation of tipping timescales are not sufficiently clear, at least for me. As written, it is difficult for readers unfamiliar with this methodology to grasp how these timescales should be understood, and what they influence. Do they represent the time for a subsystem to transition between states? If so, this should be made explicit.

In short, yes, the timescales do indeed represent the time for a subsystem to transition from one state to the other. To be clear, the collapse timescale would correspond precisely to the tipping timescale – i.e. given that an element has been triggered, how long does it take to, on average, reach its degraded state. At the fundamental level, we have made an equivalence between the transition probability (i.e., the probability to be in the other state at the next time step) and the transition timescale (once triggered, how long does it take on average to reach the other state). We will add this discussion as the 3rd paragraph of Sect. 2.3 to better clarify our treatment of timescales, which will be preceded by an extra paragraph regarding the conversion from model time into, say, years.

**1.1.3 M3: Conclusions and limitations.**

The conclusion is lengthy and could be made more concise and focused. Thank you for this feedback. To address the problem of length/focus, we will move several paragraphs from the Conclusions into the Results, namely to the end of the Sect. 3.3. Moreover we will reorder and rewrit several paragraphs to include discussions around your comments and to have a clearer

flow namely: 1) a discussion on the effects of interactions, 2) one on the effects of temperatures, 3) advancements due to and limitations of our approach, 4) future direction. In my opinion, several parts belong in the results section. More importantly, the discussion lacks sufficient reflection on limitations. Key questions remain:

– Is a fully probabilistic representation appropriate for physical systems? Is it more suitable than, say, Wunderling 2021?

We acknowledge that Wunderling et al. (2021) is more suitable to explain effects and phenomena such as critical slowing down, while explicitly including bifurcations. Thereby, it employs itself more simplified dynamics for each tipping element than the individualised models, such as the Stommel model used for AMOC (Stommel, 1961; Mehling et al., 2024). However, the model by Wunderling and colleagues conceptualises interactions deterministically as opposed to the explicitly probabilistic interactions discussed in Kriegler et al. (2009). Furthermore, it is more computationally demanding. We use a 2-state Markov chain as a higher-level abstraction of the double-fold bifurcations to gain a more computationally efficient and ultimately more interpretable model fit for a risk assessment framework. The outcomes of our model qualitatively reflect their outputs while being computationally cheaper and remaining coherent to the original expert elicitation.

– I understand that the interaction is still linear between the subsystems. This is a limitation, similar to previous work, and has to be acknowledged as such.

Thank you for raising this point, which allows us to clarify that the interactions we consider are not all linear, and in particular the probability to collapse $\tilde{p}_{c,i}$ for any given node $i$ is inversely proportional to the stabilising interactions, $a_{ji}^{-}D_j$, coming into $i$ from node $j$, as can be seen in Eq. (5). For the destabilising interactions, we agree that the collapse probability is a polynomial of order 1 of incoming destabilising interactions. However, since the map between state variables that dynamical systems models and the degraded state in our Markov chain model is non-linear, then linear interactions in probability space may not correspond to linear interactions in state variable space. We will add this discussion to the Conclusions section.

– What are the implications of limited data quality?

Beyond the perennial problems of reduced model precision and accuracy, some of the strongest implications due to limited data quality is that potentially some very key interactions are entirely missed which may have large consequences. Even for some interactions, which are 'known' about and discussed in Wunderling et al. (2024) (e.g. the interaction due to ENSO on AMOC) it is largely unclear how strong the interaction is and even whether it is stabilising or destabilising.

For the interactions we have included, as we can see in Fig. 3 some can cause an change in risk to be degraded by 10% (e.g. the effects of ENSO on the coral reefs) and even by up to almost 30% (e.g. the stabilising interaction due to AMOC on GIS). To that extent, a missing interaction could change the results by a large amount. Although this is an implication of limited data in any model – missing interactions and mechanisms – there does seem to be a particular scientific gap regarding the quantification and knowledge of inter-climate subsystem interactions.

Another major problem is how to quantify the interactions in Wunderling et al. (2024) that either were not discussed in Kriegler et al. (2009) or which has since been updated and re-evaluated by the 2024 review. Thus, in order to overcome the lack of data, we had to use the few interactions which are present in both and consistent to draw equivalences between the qualitative strength and an edge weight (details in our response to the Appendices and we will such discussion into Appendix A2), and though this process is at least somewhat systematic it is by no means unambiguous and produces room for uncertainty.

Finally, despite multitudes of Model Intercomparison Projects (MIPs) for various climate models, the fact that the Tipping Point MIP (TIPMIP) (Winkelmann et al., 2025) is at the moment beginning to publish its Tier 1 results, speaks to a lack of data and knowledge in general about even individual tipping elements. We hope that such experiments reveal more about the individual elements but also may serve as baselines to compare our results against.

– What new insights are provided beyond previous studies such as Wunderling 2021? One, we are able to quantify the effects purely due to interactions on the individual and total risk to be degraded, by comparing to an interaction-free baseline. We see that in the short term interactions are most impactful after 2100, while at equilibrium they produce the biggest effects for temperatures hotter than 1.5°C. Two, unlike Sinet et al. (2023, 2024), we find that the AMOC is ultimately destabilised by its interactions – i.e. the presence of interactions has a net effect of increasing its risk to be degraded – and moreover that it cannot prevent a global cascading event as they had postulated. We will emphasise the points earlier in the Conclusions.

– At present, many of the conclusions appear to be direct consequences of the assumptions rather than emergent results (e.g., interactions destabilise the system, when it is hypothesised that most are destabilising). It is crucial to articulate clearly what the results obtained via this framework add to current knowledge. Regarding interactions, on the whole, increasing the risks of being degraded, we agree that the result is not an emergent property or phenomenon. However, we believe it is important to note that the few strong stabilising interactions are not enough to counteract the remaining destabilising interactions. Some works have theorised for example that WAIS stabilising AMOC might be sufficient to prevent a global cascading event (Sinet et al., 2023) which our result in particular rebuts and at least draws attention to the fact that many of the other interactions need to be included.

– How far could one go in studying coupled tipping elements via this framework? Although what one can learn about the physical individual tipping elements or climate subsystem *per se* is limited, our framework can inform how to *conceptualise* TE-TE interactions and thus coherently include more climate subsystems and interactions. In particular, our model asks, following the conceptual logic of probability factors that Kriegler et al. (2009) had discussed and given our most up-to-date conceptualisation/understanding of such interactions, namely (Wunderling et al., 2024), what are the implications and how can we unify them in a mathematically coherent manner.

**1.1.4 M4: Readability.**

The manuscript contains many convoluted sentences and vague formulations, which make the first read unnecessarily difficult for a paper that is otherwise not highly technical. I strongly recommend careful editing for clarity and precision. I have listed specific examples below.

Thank you for your thorough feedback. We really appreciate your careful reading of our paper and the specific comments. We agree our manuscript had been convoluted and vague in places. To rectify this we will: move different paragraphs to improve flow of the text; remove two paragraphs entirely from the introduction to cut down on excessive tangents; added more explicit signposting in the Methods section to ease reading; and address your specific comments, see below for our detailed responses. Moreover, we will revise Table A2 in Appendix A2 to include the probability factors (PFs) evaluated by Kriegler

et al. (2009) and the maximum link strength in Wunderling et al. (2021) for improved clarity and transparency regarding edge parametrisation.

**1.2 Specific Comments**

**1.2.1 Abstract**

1: Capitalize "Atlantic Meridional Overturning Circulation". Fixed.

3: "collapse" and "into a degraded state" sound redundant; collapse alone should be enough. Fixed, we will instead write "being degraded". We wish to highlight, however, that "collapse" – i.e. the process of collapsing – is not necessarily the same as being in the degraded state. A climate subsystem might be triggered – e.g. by crossing its threshold – but the process of collapse is not instantaneous, it takes time. Therefore, a system might be undergoing collapse without being in the degraded state. Only once collapse has finished will the system be in the degraded state.

5: Not sure, but isn't "literature-based expert assessment" more correct? Both expressions "expert assessment" and "belief assessment" are accurate insofar as an assessment of expert judgement – i.e. the actual surveying of experts by Kriegler et al. (2009) – is ultimately an assessment of the beliefs of experts (Lam and Majszak, 2022). Kriegler et al. (2009) explicitly use the term belief – "[t]his points to the strength of expert elicitations in providing a holistic picture of beliefs". For clarity, however, we will use "expert assessment" instead.

6: "equilibrium risks of tipping elements" is not understandable at this point. We will replace this with "risk of being degraded for eight interacting tipping elements and ENSO at equilibrium".

6: "their stationary distribution" alone, referring to tipping elements, is vague. We will remove "stationary distributions" in favour of "this equilibrium solution", where "this" is in reference to the previous sentence in the abstract, addressed in the above point.

8: Capitalize "Shared Socio-economic Pathway". Fixed.

8: "Hypothetical" is not correct; I guess you mean "idealised". Idealised is also not quite what we mean, instead we will opt for "interaction-free baseline".

11: More correct would be "SSP scenarios with the strongest greenhouse gas emissions". Fixed.

12: What do you mean here by comprehensive? It does not seem fair to me. At least, in climate science, it is a confusing denomination. We will remove "comprehensive".

**1.2.2 Introduction**

15: "Earth systems" is not correct. I think you mean "climate subsystems". Fixed.

15–18: The second sentence is not really an example of the first. The initiation of the marine ice sheet instability beyond some tipping point is indeed a nonlinear effect, but does not have to do directly with either path dependency or chaotic behaviour. We will remove path dependency and chaotic behaviour, and instead add "threshold effects in temperature" as an example of non-linearity which then will flow better into the example of WAIS in the next sentence.

21: Please explain what you mean by "levels". We meant "levels" in the sense of for example at the "level" of socio-ecological/socio-environmental interactions, i.e. beyond pure geographical size. To avoid confusion, we will remove "levels".

25: More correct would be "most interactions between tipping elements are destabilizing". Fixed.

26: Say at least "tipping cascades". I would say that here can be a moment to explain what a tipping cascade is. Thank you for this point, we will include "tipping cascades" and add a brief descriptor of what they are namely where "one element tipping cause others to tip, potentially cascading through the entire system".

27: Replace polar ice caps by "the WAIS". I understand that the sentence does not work with this change, but the paper does not say that ice caps stabilize the AMOC. Fixed.

28: Here and in other places, "saltwater" should be "freshwater", or maybe "meltwater" if originating from an ice sheet. Apologies for this mistake, it will be rectified everywhere.

33: Replace point by , 35: I don't understand "system model". 31–44: Personally, I find that these two paragraphs are a bit of an overkill in this introduction. Or at least, they give me the feeling of taking too much space. Thank you for these points, we agree they had sounded excessive and out-of-place and so we will remove both.

55–58: There is excessive use here of the —, this part is not readable. Please separate it into sentences, as it is I do not really understand what the authors are trying to say. We will reorder the clauses and change them into full sentences.

55: "noise-induces tipping in some climate models occur when" is not grammatically correct. Fixed the typo.

56: The description of N-tipping quoted from Ashwin 2021 introduces unnecessary technicality. We will remove the direct quotation and instead have described it merely as "where fluctuations push a system outside of a basin of attraction".

57: Maybe it is because of the weird phrasing, but I don't understand "or through confidence intervals on parameter values". We will replace this with "sampling parameters from confidence intervals".

64: You do not remedy the shortcoming of linear interaction though, which is a big one when it comes to interacting climate tipping elements. As far as I understand, the interaction impacts the probability linearly. While we agree that the probability is a linear polynomial in *destabilising* interactions, the dependence on *stabilising* interactions is non-linear, namely inversely proportional. To that end, we are at least moving closer to all interactions being non-linear, as at least *some* of our interactions are indeed non-linear. Moreover, the shortcoming we are trying to remedy is not necessarily that interactions have been modelled as linear, but rather that the effect of interactions act directly on the probability to be degraded rather than a state variable. Since the "transitioned" or "baseline" regimes in Wunderling et al. (2021) are not linearly dependent on the state variable – consider that the indicator function to be in the, say, transitioned regime is essentially a step function from state variable $x_i$ and is hence non-linear – then it is not true that having probabilities which are linear in destabilising interactions (what we do) implies state variables which are linear in those interactions (what Wunderling et al. (2021) do). In the manuscript we will add further emphasis to the fact that the probability factors act on probabilities not on state variables and add a sentence clarifying this point.

77: "which accounts for the physical sensitivity of the element, or at least uncertainty in the critical temperature" is a very vague formulation, especially the "at least". We will replace this description instead by "In effect, even though there is uncertainty with regards to the exact value of the critical threshold, the higher the global warming level is above the minimum threshold

estimate, the more likely we believe each element is to begin to tip."

185  82: "including under external forcing" is awkward in this sentence. We will remove this and instead talk about including effects due to global warming.

83: see M1. Fixed.

85: "equilibrium degraded probability" sounds like the probability is degraded. I imagine there must be a better formulation. For brevity, we had often used "degraded risk" or "degraded probability" to be synonymous with "risk of being degraded" or

190  "risk to be degraded". In order to avoid confusion, we will replace all such usage with terms like "risk of being degraded" and so on.

Figure 1: In the caption, see M1. Fixed.

**1.2.3  Methods**

92: I am not very familiar with these methods, so here it could be useful to have one or two short sentences that describe what

195  will be done. Like "We describe each tipping element by . . . ". We will add a paragraph at the start of the whole section to signpost and describe what we will do with Markov chains. We will also expand the sentences at the start of Sect. 2.1 by first noting that Markov chains are characterised by its states and transition probabilities.

95: Here, I understand "sufficiently" should be "fully". In General: it is more common for me to write equations as soon as they are mentioned. Fixed.

200  102: I don't understand the notion of stability of an equation in this context. Is it the stability of the stationary distribution? Yes, the stability of the stationary distribution. We will make this more explicit.

102: Absolute value of the Jacobian can mean a lot of different things, so it should be clear here. We will clarify this to meaning finding the eigenvalues of a Jacobian matrix.

109: "the adjacency matrix". Fixed. 133: see M1. Fixed.

205  135: It is not explained what the two timescales are. In regard to M2, it would be a good moment to explain why the timescales are involved at this moment, in this way, and why in the value of the internal probabilities. As I read it, I understand that probabilities evolve with respect to these timescales, which is not what a tipping timescale is (i.e. the time for transitioning from a state to another) See our response to M2. In short, we will now add two paragraphs expanding upon our treatment of timescales.

210  144: Here, it is not indicated how you set these parameters beta. I understand it is in an Appendix; refer to it. Fixed. 148: Eq 10 appears many pages after; for me this is awkward. We will move the entire paragraph discussing Eq. 10 to just after Eq. 10 in Sect. 3.1.1. We kept the equation in the results section as this is a theoretical result.

Table 1: For the table and caption, see M1. Also, regarding the *, I did not manage to find this information in Armstrong. Finally, the index i should be explained. Fixed the point regarding ENSO. The data is found in the Supplementary Data

215  S1 of (Armstrong McKay et al., 2022) and we will add a more specific pointer to this supplementary data in the table caption. Regarding the index, we will remove it from the table, but in general $i$ will index tipping elements and ENSO, i.e. $i \in \{\mathrm{REEF}, \mathrm{ASI}, \cdots, \mathrm{GIS}\}$.

161: see M1. Fixed.

167: I imagine you refer here not to Armstrong but Kriegler? Yes, thank you for catching this mistake. We will fix this. 170: Please reformulate the first sentence. We will remove the middle clause such that the sentence will be: "A major challenge here is the deep uncertainty within the literature where interactions are concerned (Lam and Majszak, 2022; Wunderling et al., 2024)."

171: I would not use "rife", as it is not very non-English friendly. Same for dyadic, could use bilateral, or bidirectional. We will remove rife in favour of simply "within the literature". We will similarly remove "dyadic" in favour of "particular", since bilateral and bidirectional would both imply that all pairwise interactions go both ways, whereas this is not necessarily true as there are some interactions which are unidirectional, but certainly connect two nodes.

180: What do you mean all 2 interactions? Out of the original 19 interactions (i.e. edges) from Wunderling et al. (2024), 2 of them have unclear response type and unclear response strength. We will add this clarification to the text.

180–181: Do you reflect on that later? It is a good reason to justify that you cannot include it, not that it should not be. We do so in the Appendix A2 but will re-emphasise the point in the Conclusions.

**1.2.4 Results**

188: I understand that the stability is rather the object of the next subsection. We will remove the mention of stability here.

238: I did not catch why eq 10 is "far from a theoretical result"; it seems to me that it is precisely what it is. Also, "into interacting tipping elements in the Earth system" could already be more specific. Rightly or not, for some people, theoretical results are not as important or impactful as empirical results; that theoretical results are sometimes seen as some little 'neat' exercises in mathematics, done purely for the sake of theory. Here we wanted to appeal to those people, by highlighting how to interpret such results thus bridging from the mathematical results into the computational and simulations results. Regarding your latter point, we will highlight that the insights are into "the effects of interactions – following the logic of probability factors (Kriegler et al., 2009) – on the nine climate subsystems".

247–248: Seeing this figure for the first time, I would judge that this variation is fairly weak on Fig 2, given the fact that most interactions are destabilising. I would like the authors to reflect on that. At the level of total risk, i.e. Fig. 2, you are correct that the net result of interactions is fairly weak. However, there are two things to keep in mind. One, there are still some interactions which are stabilising, so this cancels out some of the destabilising effects of other interactions. Two, we see this most clearly in the individual breakdown of risks, i.e. Fig. 3, where for example the effects of interactions on GIS (see panel (g)) is a big reduction in risk particularly around 3-5$°$C. Within the 5-95 percentile range, some interaction matrices (i.e. samples from the ensemble) can even reduce the risk of GIS being in the degraded state by 30% from roughly 90% to 60%.

252: Why the use of the French word "sans", also in the figure? I think "scenario without interaction" or "interaction-free scenario" would work. We have changed all "sans interaction" to "interaction-free" in Figures and in the manuscript text.

Fig 2: Again, degraded risk sounds like it reads as if the risk is degraded. For the figure and caption, also see M1. Fixed.

259: typo for "cab" ⟶ "can". Fixed.

Fig 3: For the caption, see M1. Fixed.

Fig 4: For the caption, see M1. Fixed.

In section 3.3: these are "extended SSP scenarios", not "SSP scenarios". Fixed.

**1.2.5 Conclusion**

315: This sounds like a direct consequence of the assumptions. Maybe you mean that, considering a robust representation of interactions, global warming remains the most important factor? Yes we did, we will qualified this with "considering a robust representation of interactions". 316: see M1. Fixed.

332: What do you mean here by more complex models? In this paragraph, only Wunderling 2021 and 2023 are cited. Do you compare to these references? If yes, "more complex" is misleading. By more complex we mean with respect to our model, thus to that end both Wunderling 2021 and 2023 are more complex. For clarity and flow, we will no longer refer to them as more complex.

332: In regard to M3, it sounds trivial, as most interactions are assumed to be destabilising. We will remove this sentence and instead mention that despite the majority of interactions being destabilising, the net effect of interactions is fairly weak at the level of total risk (Fig 2).

342: saltwater should be freshwater or meltwater. Fixed.

347: Here again, not sure how Wunderling is more complex than the presented model. See above comment.

358: What does "nor the entire earth system" mean here? There is no representation of the Earth system in this model. Apologies for this mistake. We had meant Earth system in the more colloquial sense of Earth as a system i.e. the entire network of climate subsystems, as opposed to the Earth system as in Earth system models (ESMs). For clarity, we will change this to the "entire network". Note that this paragraph and the preceding one have now been moved into the Results section.

363: I guess the author means 'safe overshoot' here. Fixed. 367: see M1. Fixed. 395: I would begin with "to conclude", or similar. Fixed.

**1.2.6 Appendices**

Around 435: It seems to me that the method for choosing interaction strength is very subjective, and is not really announced as such. Thank you for this point. We have tried to be as transparent as we can in the methodology and to try to be as methodical/systematic as we can. To be clear there are effectively two parts to choosing interaction strength; one is the qualitative evaluations from literature review and the other is translating the qualitative words (e.g. "strong", "weak", etc.) into specific numerical values. It is unclear as to where you believe the methods are subjective but let us try to clarify the role of subjectivity, if any, in both these aspects.

In the former, the response strength in words stems from the extensive literature review (Wunderling et al., 2024) which, insofar as any literature review requires some level of subjective expert evaluation, contains some subjective choices. However, just because there is some subjective evaluation involved, does not mean it is necessarily bad and, as Lam and Majszak (2022) discuss, a lot of subjective beliefs of experts are used to formulate high level reports, such as the IPCC Assessment Reports.

**Table 1.** Table of interactions including the minimum and maximum possible values for their edge weights. In our work, each interaction has weight $w_{ij}$ which is uniformly distributed from a minimum weight $m_{ij}$ to a maximum weight $M_{ij}$, in other words $w_{ij} \sim U(m_{ij}, M_{ij})$. The sign of the weight corresponds to the sign of the corresponding signed edge ($a_{ij}^{\pm}$) while the magnitude of the weight is the value of the signed edge, in other words $a_{ij}^{\text{sgn}(w_{ij})} = |w_{ij}|$. For reference and comparison, we also tabulate the response type and/or strength discussed in three previous references, namely: the probability factors (PF) from Kriegler et al. (2009); the maximum link strengths that Wunderling et al. (2021) had inferred from the previous probability factors; and the updated literature-based assessment from Wunderling et al. (2024) in terms of response type and strength.

| | | Kriegler et al. (2009) | Wunderling et al. (2021) | Wunderling et al. (2024) | | | |
|---|---|---|---|---|---|---|---|
| **Source** | **Target** | **Probability factor** | **Max. link strength** | **Response** | **Strength** | $m_{ij}$ | $M_{ij}$ |
| GIS | AMOC | $(+)\,[1,10]$ | $+10$ | Destabilising | Strong | $+0.1$ | $+10$ |
| WAIS | AMOC | $(\pm)\,[0.3,3]$ | $\pm3$ | Unclear | Weak, Moderate | $-3$ | $+3$ |
| AMOC | GIS | $(-)\,[0.1,1]$ | $-10$ | Stabilising | Strong | $-10$ | $-0.1$ |
| AMOC | WAIS | $(+)\,[1,1.5]$ | $+1.5$ | Destabilising | Unclear | $+0.1$ | $+1.5$ |
| GIS | WAIS | $(+)\,[1,10]$ | $+10$ | Destabilising | Moderate | $+0.1$ | $+5$ |
| WAIS | GIS | $(+)\,[1,2]$ | $+2$ | Destabilising | Weak | $+0.1$ | $+2$ |
| AMOC | AMAZ | $(\pm)\,[0.5,4]$ | $\pm2$ up to $\pm4$ | Unclear | Moderate | -5 | $+5$ |
| AMOC | ENSO | $(+)\,[1,2]$ | – | Destabilising | Weak | $+0.1$ | $+2$ |
| ENSO | AMAZ | $(\pm)\,[0.8,1.5]$ | – | Destabilising | Strong | $+0.1$ | $+10$ |
| ENSO | WAIS | $(+)\,[1,5]$ | – | Destabilising | Weak, Moderate | $+0.1$ | $+3$ |
| ASI | AMOC | – | – | Destabilising | Weak, Moderate | $+0.1$ | $+3$ |
| AMOC | ASI | – | – | Stabilising | Weak, Moderate | -3 | -0.1 |
| ASI | GIS | – | – | Destabilising | Weak | $+0.1$ | $+2$ |
| ASI | PERM | – | – | Destabilising | Weak | $+0.1$ | $+2$ |
| ENSO | REEF | – | – | Destabilising | Strong | $+0.1$ | $+10$ |
| AMOC | WAM | – | – | Destabilising | Unclear | $+0.1$ | $+1.5$ |
| PERM | AMOC | – | – | Destabilising | Unclear | $+0.1$ | $+1.5$ |

Regarding the translation of qualitative terms for response strength in Wunderling et al. (2024) into edge weights in our model, we tried to stick to the methods of Wunderling et al. (2021) as closely as possible and where possible. We have revised Table A2 (see below for the new version) in Appendix A2 to include the probability factors from Kriegler et al. (2009) and Wunderling et al. (2021) in order to be more transparent in our methodology.

For some qualitative strengths there are clear, consistent correspondence between probability factor value and qualitative strength, while others have been revised. Take the GIS-AMOC interaction as an example: Kriegler et al. (2009) found it had a maximum probability factor of 10, meaning the probability to collapse of AMOC increases by a factor of 10 if GIS collapses, while Wunderling et al. (2024) evaluated the interaction as strong and destabilising. On the other hand, the GIS-WAIS interaction which Kriegler et al. (2009), and therefore Wunderling et al. (2021), was considered strong with a maximum

probability factor of 10, was revised in the later work of Wunderling et al. (2024) to only being of moderate, though still destabilising, effect.

To this end, we use the few interactions which are consistent throughout all three references, to make an equivalence between the qualitative strength to edge weight in our model. For example, AMOC-GIS corresponded to a PF of 0.1, in other words the probability to collapse of GIS is *reduced* by a factor of 10 given that AMOC has tipped. Wunderling et al. (2024) also evaluated this as strong and stabilising. Therefore we make the equivalence that a "strong" interaction would have have a maximum absolute edge weight of 10. In doing so, we have tried to limit adding our own subjective belief of individual interaction strengths, by utilising this more systematic method. We will add this discussion into Appendix A2.

In general: I have seen only references to appendices A1 and C. Please refer to appendices when relevant. Fixed.

**1.2.7  Suggestions**

3: I would write "In doing so, . . . ". Fixed.

4: I would write "therefore, in this work . . . ". Fixed.

9: Here I would write ". . . . For instance, ". Fixed.

23: Maybe "each other". Fixed.

24: I would change ripple for propagate. Fixed.

150: I would write "therefore, for mixed terms, . . . ". Fixed.

162: It would be clearer as "sensitivity to . . . ". Fixed.

163: I would write "for the sensitivity". Fixed.

166: I would rather write "although this review dealt with describing response strengths in qualitative terms . . . ". Fixed.

180: I would write "Moreover, note that, in this work, . . . ". Fixed.

**Reviewer 2**

**General Comments**

I generally think this is an interesting paper. It is well within the scope of the journal. The approach seems novel, and to be build on appropriately cited literature. However, there should be better clarity in the conclusion as to exactly what is novel, and how substantial the contributions of this paper are - this is currently a little unclear. The assumptions informing the model are generally valid, although in a few places could be laid out better, the methods should be better justified, and the extent to which the conclusions are dependent on particular uncertain parameter values/ranges should be laid out in more depth. Similarly, the limitations of the model created could be better justified. First, thank you for your comments and feedback overall. Second, with respect to the conclusion we agree that it had been rather unfocused and lacked more in depth discussion regarding model limitations. To this end we will reorganise much of the Conclusions while adding several discussion points such as model limitations (suitability of probabilistic methods, linearity of destabilising interactions, the lengths one could go studying

coupled tipping elements via this framework), uncertainty regarding interactions, temperature independence of recovery (see our response to your Methods comments), etc. (for more details, please see our response to Reviewer 1's M3 comment).

325    I have a number of key specific issues with the paper that I think should be addressed (Specific Comments), and less significant suggestions/concerns which I leave addressing up to the authors discretion (Minor Comments).

**Major Comments**

**Abstract**

The abstract suggests the paper presents a "comprehensive risk assessment framework". The paper does not do this, and should
330    be revised to suggest this. Fixed, we will remove the word "comprehensive".

**Methods**

These are my overall comments regarding the model itself.

In your methods section, you do not have the recovery probability as influenced by GMST. However, recovery clearly is influenced by GMST, and this is especially relevant if we are either dealing with overshoot scenarios (which are discussed later
335    in the paper with SSP1-1.9) or scenarios where certain forms of Solar Radiation Modification are used. One justification is that timescales can be long enough that we can think of the tipping elements as irreversible over human relevant timescales, and this appears to be the justification in Appendix A and Table 1. However, this isn't correct for all the tipping elements, as noted about REEF in the paper. For other tipping elements, it isn't implausible that they have decadal recovery times, even if you have assumed they have longer recovery times in this paper. Not having any GMST dependence for recovery probability
340    reduces the utility of your model if you use different parameters for recovery time such that it can't be ignored. To me therefore, at the very least you need to justify why you don't have the recovery probability influenced by the GMST in the main body of the text. Likely ideally, you would actually incorporate GMST dependence.

Thank you for this point, you are absolutely correct that very likely recovery is dependent on temperature and thus the lack thereof is a limitation of our model. Although we have an explicit temperature-dependence for collapse, we do not do so for
345    recovery, due to the deep uncertainty within the literature. In some climate subsystems, such as the Greenland and Antarctic ice sheets, strong hysteresis (Armstrong McKay et al., 2022; Lenton et al., 2025) mean that any threshold effects for recovery may not occur at the same temperature value as the corresponding collapse threshold. In others, even the question of reversibility is still under debate, let alone having a fully quantified temperature-dependence. For example, while Lenton et al. (2025) note a growing amount of evidence to suggest Arctic sea ice loss is irreversible, they find it is not yet sufficient to assess it as such.
350    Therefore, we try to remain as agnostic as possible with respect to the temperature-dependence of recovery, minimising the amount of additional assumptions, by assuming temperature *independence*, in this paper. We suspect in the short-term (multi-centennial scales) that temperature-dependent recovery will not change our results greatly, given the extra order of magnitude in the recovery timescales, however the effects in the long term and at equilibrium may be large – dominating any effects due to interactions – and warrants further detailed investigation. We will add this discussion to the Conclusions.

355     I also think it's important to discuss what role GMST is playing in this model more broadly. In general, in the tipping liter-
ature, anthropogenic drivers of tipping tends to be collapsed into a discussion of GMST and critical (temperature) thresholds.
However, for a number of tipping points, especially Amazon Dieback and Coral Reefs, there are other anthropogenic drivers
that are not correlated with GMST. Similarly, for most other tipping points, the drivers are caused by GMST rise, but aren't
necessarily entirely linked. If we could decouple GMST rise from the rest of the climate system, as an intervention like Solar
360 radiation Modification could do, this would change a number of these factors. I think, therefore, noting the exact role that
GMST is playing in the model would be helpful. There are a two clear ways you can do this. The first is that you can simply
clarify this as a limitation of your model. The second is that you can argue that GMST is a useful proxy for all anthropogenic
influence (since it is currently by a very long margin the most dominant - and likely would be even under SRM deployment),
and include all the other complexity under your beta term. Since the uncertainty is so large, my guess is this is simply dwarfed
365 by your already existent uncertainty, so would not change your results. Thank you for raising this point and your suggestions.
We will add that we use global warming since it is a useful proxy and in many cases it dominates other drivers.

**Results**

The presentation of the results of the "short term" case is misleading by the choice of the two dates to discuss: 2100 and 2350.
Looking at Figure 4, it is clear that much of the increase in the tipped probability across the tipping elements happens in the
370 2100s, which means the choice of these two dates might give the impression that the tipping will happen later than it in fact
does. I know 2100 is a very common date used in the climate literature, and therefore I might use three dates in the discussion:
2100, 2150 and 2350 for example, to illustrate this massive increase in tipping probability soon after 2100. We will add an
extra paragraph discussing the effects of interactions (previously Fig. S3 now Fig. 6) which also discusses 2150. We have also
added reference to 2150 in other areas of the results.

375     You almost exclusively discuss the arguably two least likely projections, SSP 1-1.9 and SSP 5-8.5. At least a small amount
of discussion of the more "middle of the road" scenarios would be helpful to give the readers a more complete picture. We
will add more explicit mention of SSP2-4.5, in particular with respect to the minor decrease in risk for the coral reefs post
2100-2150 and the slightly stronger increase for AMOC and the Amazon. We also have discussed SSP3-7.0 frequently, as it is
often very similar to SSP5-8.5.

380     It seems very strange to me that S3 is not in the main paper. This seems to be one of the key results for the paper. I also think
this needs to be discussed in a fair amount of depth in the results or conclusions, as its a really important take away. This better
comparison to the no-interactions case seems particularly important given your mentioning of a similar point in the abstract.
We will move Fig. S3 from the supplementary into the main text, as future Fig. 6 and will added a paragraph discussing these
results, also highlighting that interactions are strongest in 2150-2200.

385 **Conclusion**

I think I would like considerably more discussion on the limitations of your approach, which seem only minorly discussed.
This will be important to how people should interpret your results, since it is such a simplified model. Moreover, this will be

helpful for people building on your results. Thanks for this point, we will move some paragraphs from the Conclusions into the Results and will add several paragraphs discussing the limitations of our model, namely: lack of data regarding interactions and our efforts to accommodate this, the suitability of a probabilistic approach to physical systems (see our response to Reviewer 1's M3), and the temperature-independence of our recovery rates.

Like all models, your model is, in large part, a consequence of the assumptions going into it. I however, don't know the degree to which your results are dependent on your parameters, which are highly uncertain, so some discussion of this would be very helpful. Moreover, I'd like to get a sense if any results were surprising given the inputs used, and whether such a simplified set up has the capacity to provide novel explanatory power. This is all unclear to me, and at least should be gestured towards in the conclusion. See our response to Reviewer 1's M3 comment, but in short we will now add extra discussion in the Conclusions outlining some of the limitations of our model and the implications of lack of data both for individual tipping elements and importantly interactions between them. Moreover, following your comments above, we will also add a paragraph discussing the temperature-dependence of recovery and how that may impact the short-term and at equilibrium results.

**Minor Comments**

Introduction - please define tipping element Fixed.

Methods/Discussion: I'd like a bit more explanation as to the role of timescales of tipping and how these play a role in tipping interactions. This seems to be a limitation of your model, albeit currently somewhat small given high uncertainty. As you model this as a 2-step Markov process, the timescale is just the timescale it takes to tip from one state to another. There is no intermediate state. However, intermediate states do exist (even if tipping processes are at this stage self-perpetuating), and its not clear at which stage in the process of tipping the interactions are most prominent. This may mean the timescale of tipping underestimates when interactions can occur. Of course, this is subsumed under the other uncertainties already present in your model, but is probably worth briefly discussing. We will add two extra paragraphs into the Methods section to expand upon and explain better the role of timescales; see also our response to Reviewer 1's "M2: Timescales of tipping" comment above. With respect to intermediate states, this is certainly an important avenue to pursue – for example Mehling et al. (2024) found that long chaotic transients through the intermediate edge states for AMOC might allow some overshooting of the tipping point. Equivalently we agree that this likely means the tipping timescales are in actuality longer than previous estimates. We will add this into the Conclusions.

Lines 38-45: Whilst I agree with the positives of expert elicitation, I still think some discussion of the serious problems with it in this context would be informative to readers. Namely, very poor process understanding and a lack of useful historical observations, whilst weighing heavily against the use of GCMs, also suggest a lack of reliability for expert elicitation. Following the comments from Reviewer 1, we have removed this paragraph in order to focus and streamline the introduction more. That being said, we will add a sentence into the Discussion to include both the pros of expert elicitation that we had already written about as well as the cons that you mentioned.

420     Line 286: I think it should be stressed here that "short term" risks are still, in this paper, seen as multi-centennial. This will be good to push back against some of the more catastrophist thinking about tipping elements. We will add "(up to multi-centennial times)" to highlight that short term is not just on the scale of, say, decades.

    Figure 4 should be edited for clarity. Namely:

      – You need to mention that not all the scales are the same for each of the graphs Fixed.

425       – The scales are not actually quite the same even for a to f, which all go from 0-100% on the y axis. Please edit them so they all go only from 0-100% (some go above 100%), and that they are all scaled the same Fixed, moreover we will also make panel (i) also go from 0-100%.

      – I should have the same scale as a to f. I don't really think there is a good justification for why it stops at roughly 95% on the y axis. Fixed.

430     Line 380-384: You discuss exogenous stabilisation of particular tipping elements. I think it is useful to not skirt around the subject, and explicitly state what you mean here. Whilst there are local geoengineering interventions, these are fairly nascent. I suspect this is mostly referring to either localised or global Solar Radiation Modification. If this is so, then I would make this explicit, and cite some of the emerging analysis of how SRM interacts with different tipping elements. One interesting point here, that is the flip side of the point you raise, is that it is feasible that global SRM reduces the risk of tipping for many

435 tipping elements, whilst leaving a few without much risk reduction. AMOC looks amongst the most uncertain tipping elements as to whether SRM can reduce its risk of tipping or not, which is an interesting counterpoint to the discussion. It is not clear to me a more extensive discussion of such exogenous stabilisation is needed in the paper, but given the (useful) discussion, I think a few more lines to give the reader more context on the state of the literature on that would be helpful. Thank you for these points. In truth, we did not have any specific policy nor piece of technology, geoengineering or otherwise, in mind. Given

440 the conceptual/abstract nature of our work and in particular given some of Reviewer 1's comments (see M3 above) regarding the lack of focus in the Conclusions, we do not wish to over-extend into discussions of specific technologies, and as such will remove "exogenously stabilise" and will emphasise that we mean intervention at least at the level of the stylised network model.

    Line 385: Update citation to reflect the publication of the 2025 Global Tipping Point Report Fixed.

**References**

Armstrong McKay, D. I., Staal, A., Abrams, J. F., Winkelmann, R., Sakschewski, B., Loriani, S., Fetzer, I., Cornell, S. E., Rockström, J., and Lenton, T. M.: Exceeding 1.5°C global warming could trigger multiple climate tipping points, Science, 377, eabn7950, https://doi.org/10.1126/science.abn7950, 2022.

Kriegler, E., Hall, J. W., Held, H., Dawson, R., and Schellnhuber, H. J.: Imprecise probability assessment of tipping points in the climate system, Proceedings of the National Academy of Sciences, 106, 5041–5046, https://doi.org/10.1073/pnas.0809117106, 2009.

Lam, V. and Majszak, M. M.: Climate tipping points and expert judgment, WIREs Climate Change, 13, e805, https://doi.org/https://doi.org/10.1002/wcc.805, 2022.

Lenton, T. M., Milkoreit, M., Willcock, S., Abrams, J., Armstrong McKay, D., Buxton, J., Donges, J., Loriani, S., Wunderling, N., Alkemade, F., Barrett, M., Constantino, S., Powell, T., S. S., Boulton, C. A., Pinho, P., and Dijkstra, H.: The Global Tipping Points Report 2025, https://global-tipping-points.org/, 2025.

Mehling, O., Börner, R., and Lucarini, V.: Limits to predictability of the asymptotic state of the Atlantic Meridional Overturning Circulation in a conceptual climate model, Physica D: Nonlinear Phenomena, 459, 134 043, https://doi.org/https://doi.org/10.1016/j.physd.2023.134043, 2024.

Sinet, S., von der Heydt, A. S., and Dijkstra, H. A.: AMOC Stabilization Under the Interaction With Tipping Polar Ice Sheets, Geophysical Research Letters, 50, e2022GL100 305, https://doi.org/https://doi.org/10.1029/2022GL100305, e2022GL100305 2022GL100305, 2023.

Sinet, S., Ashwin, P., von der Heydt, A. S., and Dijkstra, H. A.: AMOC stability amid tipping ice sheets: the crucial role of rate and noise, Earth System Dynamics, 15, 859–873, https://doi.org/10.5194/esd-15-859-2024, 2024.

Stommel, H.: Thermohaline Convection with Two Stable Regimes of Flow, Tellus, 13, 224–230, https://doi.org/https://doi.org/10.1111/j.2153-3490.1961.tb00079.x, 1961.

Winkelmann, R., Dennis, D. P., Donges, J. F., Loriani, S., Klose, A. K., Abrams, J. F., Alvarez-Solas, J., Albrecht, T., Armstrong McKay, D., Bathiany, S., Blasco Navarro, J., Brovkin, V., Burke, E., Danabasoglu, G., Donner, R. V., Drüke, M., Georgievski, G., Goelzer, H., Harper, A. B., Hegerl, G., Hirota, M., Hu, A., Jackson, L. C., Jones, C., Kim, H., Koenigk, T., Lawrence, P., Lenton, T. M., Liddy, H., Licón-Saláiz, J., Menthon, M., Montoya, M., Nitzbon, J., Nowicki, S., Otto-Bliesner, B., Pausata, F., Rahmstorf, S., Ramin, K., Robinson, A., Rockström, J., Romanou, A., Sakschewski, B., Schädel, C., Sherwood, S., Smith, R. S., Steinert, N. J., Swingedouw, D., Willeit, M., Weijer, W., Wood, R., Wyser, K., and Yang, S.: The Tipping Points Modelling Intercomparison Project (TIPMIP): Assessing tipping point risks in the Earth system, EGUsphere, 2025, 1–52, https://doi.org/10.5194/egusphere-2025-1899, 2025.

Wunderling, N., Donges, J. F., Kurths, J., and Winkelmann, R.: Interacting tipping elements increase risk of climate domino effects under global warming, Earth System Dynamics, 12, 601–619, https://doi.org/10.5194/esd-12-601-2021, 2021.

Wunderling, N., von der Heydt, A. S., Aksenov, Y., Barker, S., Bastiaansen, R., Brovkin, V., Brunetti, M., Couplet, V., Kleinen, T., Lear, C. H., Lohmann, J., Roman-Cuesta, R. M., Sinet, S., Swingedouw, D., Winkelmann, R., Anand, P., Barichivich, J., Bathiany, S., Baudena, M., Bruun, J. T., Chiessi, C. M., Coxall, H. K., Docquier, D., Donges, J. F., Falkena, S. K. J., Klose, A. K., Obura, D., Rocha, J., Rynders, S., Steinert, N. J., and Willeit, M.: Climate tipping point interactions and cascades: a review, Earth System Dynamics, 15, 41–74, https://doi.org/10.5194/esd-15-41-2024, 2024.